# Flash-VAED: Plug-and-Play VAE Decoders for Efficient Video Generation

**Lunjie Zhu** [1] **Yushi Huang** [1] **Xingtong Ge** [1] **Yufei Xue** [1] **Zhening Liu** [1] **Yumeng Zhang** [1] **Zehong Lin** [2] **Jun Zhang** [1]

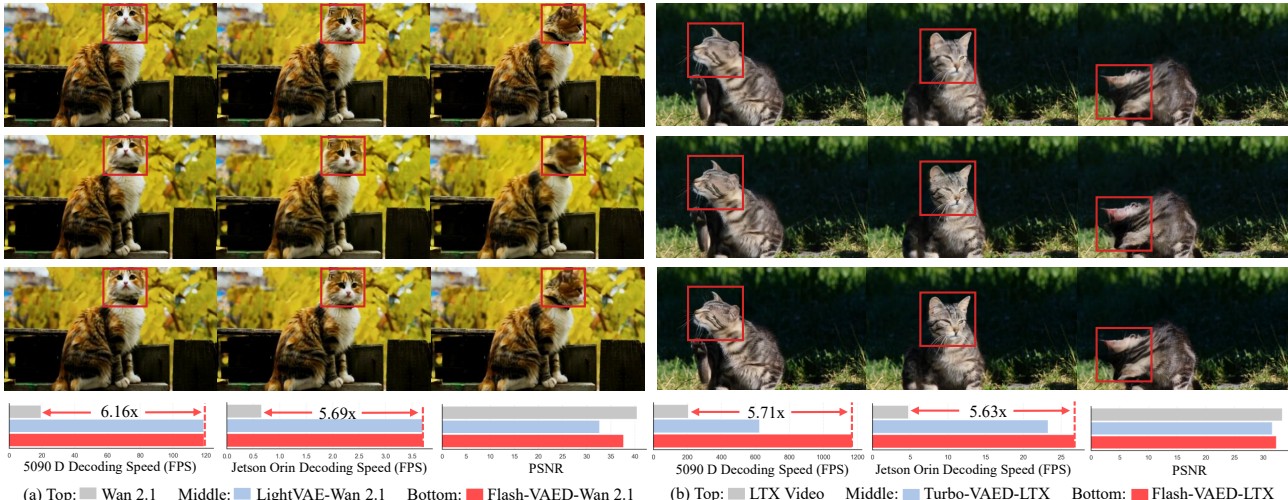

*Figure 1.* **Qualitative and quantitative comparisons of video reconstruction results.** We evaluate Flash-VAED (Bottom) against the original VAE decoder (Top) and the current state-of-the-art baseline (Middle). Flash-VAED offers the fastest decoding speed with minimal loss of fidelity to the original VAE decoder.

## Abstract

Latent diffusion models have enabled high-quality video synthesis, yet their inference remains costly and time-consuming. As diffusion transformers become increasingly efficient, the latency bottleneck inevitably shifts to VAE decoders. To reduce their latency while maintaining quality, we propose a universal acceleration framework for VAE decoders that preserves full alignment with the original latent distribution. Specifically, we propose (1) an *independence-aware channel pruning* method to effectively mitigate severe channel redundancy, and (2) a *stage-wise dominant operator optimization* strategy to address the high inference cost of the widely used causal 3D convolutions in VAE decoders. Based on these innovations, we construct a **Flash-VAED** family. Moreover, we design a *three-phase dynamic distillation* framework that efficiently transfers the capabilities of the original VAE decoder to Flash-VAED. Extensive experiments on Wan and LTX-Video VAE decoders demonstrate that our method outperforms baselines in both quality and speed, achieving approximately a **6×** **speedup** while maintaining the reconstruction performance up to **96.9%**. Notably, Flash-VAED accelerates the end-to-end generation pipeline by up to **36%** with negligible quality drops on VBench-2.0. Our code is available at `https://github.com/Aoko955/Flash-VAED`.

[1]iComAI Lab, Hong Kong University of Science and Technology, Hong Kong SAR, China [2]School of Data Science, Lingnan University, Hong Kong SAR, China. Correspondence to: Jun Zhang <eejzhang@ust.hk>.

*Proceedings of the 43rd International Conference on Machine Learning*, Seoul, South Korea. PMLR 306, 2026. Copyright 2026 by the author(s).

## 1. Introduction

Recently, artificial intelligence-generated content has witnessed remarkable breakthroughs across various domains, including text (Touvron et al., 2023; DeepSeek-AI et al., 2025), image (Labs, 2024; Xie et al., 2024), and video syn-

thesis (HaCohen et al., 2024; Wan et al., 2025), achieving unprecedented levels of realism and coherence. The success of modern video generation models is largely attributed to powerful latent diffusion models (LDMs) (Ma et al., 2025), which typically comprise video variational auto-encoders (VAEs) (Kingma & Welling, 2013) and diffusion transformers (DiT) (Peebles & Xie, 2023).

Despite their exceptional performance, video generation models are computationally expensive and slow in inference. For instance, it requires more than 30 minutes and 50 GB of GPU memory for the Wan 14B model to generate a 10-second 720p video clip on a standard H100 GPU. Such inefficiency significantly hinders practical deployment (Wu et al., 2025; Kim et al., 2025) and calls for more efficient video generation pipelines. Prior research has primarily focused on the DiT module, aiming to accelerate video generation by reducing denoising latency (Huang et al., 2026; Zhang et al., 2025) or applying model compression (Guo et al., 2020; Wang et al., 2024; Huang et al., 2025b; Huang et al., 2025c). Nevertheless, as DiT acceleration techniques advance, the latency bottleneck has gradually shifted towards the VAE decoder, which has been largely overlooked.

In light of this issue, some recent studies have explored training lightweight VAEs from scratch (Cheng & Yuan, 2025; Li et al., 2025b). However, these models often suffer from latent distribution misalignment with the original generation pipeline, incurring expensive fine-tuning on the DiT. In addition, some attempts have been made to structurally optimize the original VAE decoder (Zou et al., 2026; Contributors, 2025), yet they have neither fully investigated the underlying causes of the decoding latency bottleneck nor achieved an optimal trade-off between speed and quality.

In this work, we propose **Flash-VAED**, a family of efficient VAE decoders designed to accelerate video generation while preserving full alignment with the original latent distribution. Specifically, after a thorough analysis of the VAE decoder, we identify two primary factors that make it a latency bottleneck. First, through a singular value decomposition (SVD) analysis, we find that retaining merely ∼22% of the channels is sufficient to explain 99% of the variance in the full channel feature maps, which indicates severe channel redundancy. Based on this insight, we introduce an *independence-aware channel pruning* method that selectively optimizes a limited number of retained channels to linearly reconstruct the full channel feature maps. Second, we notice that causal 3D convolutions (CausalConv3D) dominate the latency within each decoder block. To address this issue, we analyze CausalConv3D across different stages and propose a *stage-wise dominant operator optimization* strategy, which replaces CausalConv3D with more efficient operators tailored to stage-specific characteristics. Moreover, to efficiently transfer the capabilities of the original

VAE decoder to Flash-VAED, we develop a *three-phase dynamic distillation training framework*, achieving stage-wise alignment of Flash-VAED with the original VAE decoder.

In summary, our work aims to accelerate latent decoding for video generation and makes the following contributions:

- We propose an independence-aware channel pruning method, which reduces the channel count to 12.5% − 25% of the original with minimal quality loss.
- We introduce a stage-wise dominant operator optimization strategy that targets the dominant CausalConv3D, substituting it with efficient operators based on stage-specific characteristics to maximize efficiency.
- We present a three-phase dynamic distillation training framework that enables Flash-VAED to efficiently inherit the capabilities of the original VAE decoder.
- Extensive experiments on Wan and LTX-Video VAE decoders demonstrate that our method outperforms baselines in both quality and speed, achieving an approximate **6×** **speedup** while maintaining up to **96.9% performance** in reconstruction quality. Moreover, our method accelerates the entire generation pipeline by up to **36%** with negligible quality drops on VBench-2.0.

## 2. Related Work

**Latent Diffusion Models.** Diffusion models have achieved remarkable success in generative tasks, delivering unprecedented performance in video generation (HaCohen et al., 2024; Wan et al., 2025). Among recent advancements, architectures that employ DiT (Peebles & Xie, 2023) as the backbone have become particularly prominent, demonstrating superior scalability and generation quality compared to traditional U-Net-based designs (Ronneberger et al., 2015). However, performing the iterative diffusion process directly in the high-dimensional pixel space incurs prohibitive computational costs. To address this, VAEs have been introduced to compress raw data into compact low-dimensional latent representations. This paradigm, referred to as LDMs (Ma et al., 2025), significantly reduces inference complexity while maintaining high-fidelity synthesis. Nevertheless, as models scale up, both computational overhead and inference latency escalate substantially. To address this challenge, prior research has primarily targeted the DiT module, with strategies generally falling into two categories: reducing the number of denoising steps and reducing the latency per step. The former largely employs distillation techniques (Yin et al., 2024; Ge et al., 2025), while the latter mainly utilizes model compression methods such as pruning (Guo et al., 2020; Wang et al., 2024) and quantization (Shang et al., 2023; Huang et al., 2024; Huang et al., 2025c). Yet, as the efficiency of DiT improves, the latency bottleneck has gradually shifted towards the VAE decoder. For instance, when generating an 81-frame 480p video using the original

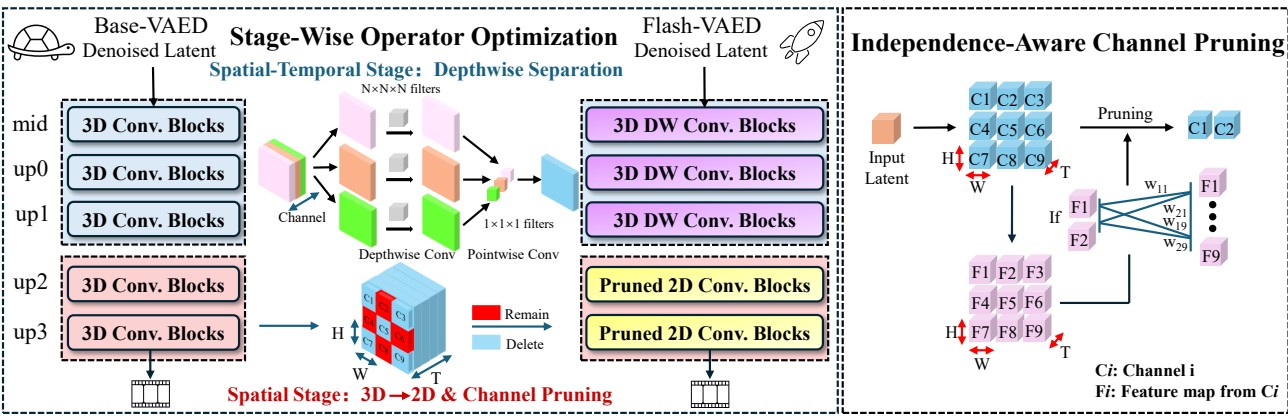

*Figure 2.* Overview of the Flash-VAED architecture[2]. The proposed stage-wise dominant operator optimization substitutes CausalConv3D with stage-specific efficient operators (left), tailored to each decoding stage. Moreover, the independence-aware channel pruning method (right) reduces the channel count to $12.5\% - 25\%$ of the original with minimal quality loss, leveraging channel independence.

pipeline for Wan 1.3B (Wan et al., 2025), VAE decoding accounts for only 2.3% of the total latency. However, with autoregressive few-step distillation (Huang et al., 2026), this proportion surges to 29.6%, representing more than a ten-fold increase.

**Video VAE Decoding Acceleration.** Modern video VAE decoders (HaCohen et al., 2024; Kong et al., 2024; Wan et al., 2025) prioritize high-fidelity and consistent decoding by employing computationally intensive CausalConv3D and a large number of channels. While beneficial for quality, these designs incur prohibitive computational and memory costs, hindering real-time deployment. To address these challenges, some studies focus on designing novel lightweight VAE architectures from scratch (Cheng & Yuan, 2025; Li et al., 2025b). However, these approaches inevitably cause a latent distribution mismatch between the VAE decoder and the pre-trained DiT. Integrating them into the generation pipeline necessitates fine-tuning of the DiT, which incurs prohibitive computational costs. Alternatively, another line of research attempts to perform structural optimizations directly on the original VAE decoder to maintain latent consistency (Zou et al., 2026; Contributors, 2025). However, these methods neither fully address the root causes of the latency bottleneck in VAE decoders nor achieve an optimal trade-off between speed and quality. For instance, Turbo-VAED (Zou et al., 2026) addresses the parameter redundancy of CausalConv3D in low-resolution layers but fails to resolve the significant inference latency in high-resolution layers. Similarly, LightVAE (Contributors, 2025) identifies channel redundancy yet fails to effectively reduce it without incurring quality loss. In contrast, our work comprehensively identifies the underlying causes of the decoding latency bottleneck, achieving substantial acceleration with negligible quality degradation.

[2]We take Flash-VAED-Wan 2.1 as an example.

# 3. Method

In this section, we analyze the two primary factors that contribute to the latency bottleneck in the VAE decoder for video generation and present our corresponding solutions. The first factor is *severe channel redundancy*, which we address by proposing independence-aware channel pruning, as illustrated on the right side of Figure 2 and detailed in Section 3.1. The second factor is the *ubiquitous presence and high inference cost of CausalConv3D*. To address this issue, we introduce a stage-wise dominant operator optimization strategy, as shown on the left side of Figure 2, with further details provided in Section 3.2. Building on these optimizations, we construct the **Flash-VAED** family. Finally, to efficiently transfer the capabilities of the original VAE decoder to Flash-VAED and seamlessly embed Flash-VAED into the original generation pipeline, we design a comprehensive three-stage feature distillation training framework, which is elaborated in Section 3.3.

For clarity, we standardize the naming of the decoder blocks. From input to output, corresponding to low to high resolutions, we refer to the main blocks as the Middle Block, Upsample Block 0, Upsample Block 1, Upsample Block 2, and Upsample Block 3, denoted by mid, $up_0$, $up_1$, $up_2$, and $up_3$, respectively.

## 3.1. Independence-Aware Channel Pruning

Channel pruning is a widely used neural network acceleration technique (Liu et al., 2017; Cheng et al., 2024). However, applying conventional pruning methods to video VAE decoders faces several challenges. First, our visualization of the output feature maps and the calculation of the cosine similarity against Channel 0, as shown in Figure 3, reveal observable redundancy. Nevertheless, pairwise similarity measures fail to provide sufficient evidence for pruning de-

cisions. Second, the extensive use of cascaded residual blocks in the decoder complicates the pruning process, as maintaining identical channel indices across different blocks is critical to preserve the model's original connectivity. A detailed explanation is provided in Appendix A.2.

To address these challenges, we shift from pairwise similarity to linear dependence, where a channel is considered redundant if it can be linearly represented by other channels. To validate this concept, we conduct an SVD analysis on the matrix of feature maps across all channels. As illustrated in Figure 4, we find that retaining only **22%** of the singular values is sufficient to explain **99%** of the total variance. Based on this insight, we propose an independence-aware channel pruning method that reduces the channel count to $12.5\%-25\%$ of the original size with minimal quality loss, while maintaining the internal continuity of the model.

Mathematically, we use $\mathbf{X}$ to denote the retained channel features and linearly reconstruct the full channel features $\mathbf{Y}$ as follows (Luo et al., 2017):

$$\hat{\mathbf{Y}} = \mathbf{W}\mathbf{X}, \tag{1}$$

where the projection matrix $\mathbf{W}$ is derived using the least squares method (Björck, 1990). Since traditional metrics for quantifying the reconstruction fidelity, such as mean squared error, exhibit inherent sensitivity to feature magnitude variations, we instead employ the coefficient of determination $R^2$ to measure how well the retained channel features explain the original full channel features (Lai et al., 2018a):

$$R^2 = 1 - \frac{SS_{res}}{SS_{tot}} = 1 - \frac{\|\mathbf{Y} - \mathbf{W}\mathbf{X}\|_F^2}{\|\mathbf{Y} - \overline{\mathbf{Y}}\|_F^2}, \tag{2}$$

where $SS_{res}$ and $SS_{tot}$ denote the residual and total sum of squares, respectively, and $\|\cdot\|_F$ is the Frobenius norm.

Based on this foundation, our independence-aware channel pruning framework unfolds in three integrated steps. First, we employ a greedy channel selection strategy to iteratively identify the optimal subset of channels. Specifically, given the current subset, the next channel is selected by ensuring that each addition maximizes the marginal gain of $R^2$ until the target number of channels is reached. Second, to ensure that the selected channels adequately represent the full feature map, we introduce a pre-pruning retained channel enhancement. We encourage the model to maximize the expressive power of the retained channels by incorporating an expressivity loss during training with respect to $\mathbf{W}$ and $\mathbf{X}$, which is formulated as:

$$L_{ce} = 1 - R^2 = \frac{\|\mathbf{Y} - \mathbf{W}\mathbf{X}\|_F^2}{\|\mathbf{Y} - \overline{\mathbf{Y}}\|_F^2}. \tag{3}$$

As shown in Figure 5, this operation significantly enhances the reconstruction capability, with $R^2$ scores improving

from an initial range of $0.87-0.93$ to consistently exceeding 0.98 across all layers. Finally, to resolve the disruption of residual block continuity caused by mismatched retained indices across blocks, we implement a topology-preserving shortcut injection. Specifically, standard identity shortcuts are replaced with $1 \times 1$ convolutions. Since retained channels are capable of linearly representing any channel within the current block, we can directly derive the mapping matrix $\mathbf{W}$ from the retained indices of the current block to the retained indices of the subsequent block using the least squares method. By initializing the $1 \times 1$ convolution shortcut of the subsequent block with this matrix $\mathbf{W}$, we effectively preserve the internal continuity of the pruned VAE decoder. More details are provided in Appendix A.2.

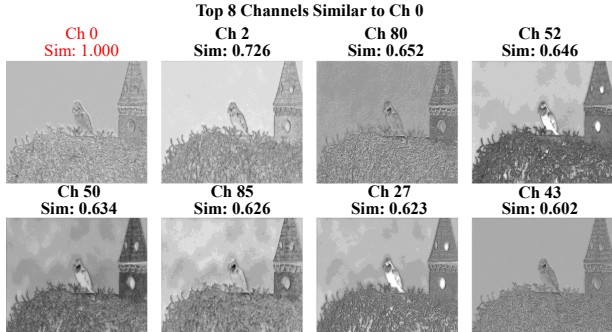

*Figure 3.* **Channel-wise similarity analysis.** We visualize the top-8 channels most similar to Channel 0. Although the feature maps exhibit visual similarity, the quantitative similarity scores are **not high enough** to support pruning.

## 3.2. Stage-Wise Dominant Operator Optimization

In addition to channel redundancy, our comprehensive latency analysis identifies CausalConv3D as another key computational bottleneck within the VAE decoder. As shown in Figure 6, this operator consistently consumes over **60%** of the inference time across most blocks, with computational costs escalating dramatically in high-resolution stages. Therefore, we characterize CausalConv3D as the dominant operator. To tackle this, we propose a stage-wise replacement strategy that substitutes CausalConv3D with tailored efficient operators based on the specific characteristics of each decoding stage. This approach achieves significant reductions in both parameter count and inference latency with negligible quality degradation.

Specifically, guided by insights from previous works (Zou et al., 2026; Sandler et al., 2018; Howard et al., 2017), we start from the deep layers (low-resolution layers). In these layers, we substitute CausalConv3D with 3D depthwise separable convolutions (3D DW Conv.), reducing the parameter count to approximately 20% of the original VAE decoder with minimal quality degradation. The theoretical analysis is provided in Appendix A.1. For the shallow

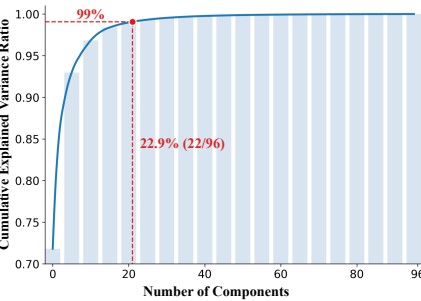

*Figure 4.* **SVD analysis on channel features.** The curve for cumulative explained variance ratio reveals the intrinsic low-rank nature of the feature maps. Notably, only 22 components (22.9% of the total) are required to explain 99% of the feature variance, providing strong empirical support for our pruning method.

layers (high-resolution layers), since temporal upsampling has been completed in deep layers, we hypothesize that the reliance on inter-frame temporal dependencies diminishes here. Consequently, computationally expensive 3D operations can be safely replaced by spatial-only operators. To validate this hypothesis, we conduct an experiment by progressively replacing CausalConv3D with 2D convolutions from deep to shallow layers. As illustrated in Figure 7, a clear trend emerges: as the replacement shifts toward higher-resolution layers, the latency increasingly reduces, while the quality loss diminishes. These empirical results strongly support our strategy to replace CausalConv3D in the shallow layers with lightweight 2D convolutions for faster inference.

### 3.3. Training Strategy: Three-Phase Dynamic Distillation Framework

To enable **Flash-VAED** to efficiently inherit the capabilities of the original VAE decoder, we incorporate feature distillation into our training objective. Following the paradigm established in prior works (Bai et al., 2023; Doshi & Kim, 2024), we define a feature-based distillation loss as follows:

$$L_{\text{distill}} = \sum_l \frac{1}{\text{numel}(f_l^F)} \sum_i \left\| \sigma(f_l^F)_i - f_{l,i}^O \right\|_1,$$

$$\text{with} \quad \sigma(\cdot) = \begin{cases} \text{Identity}(\cdot), & \text{phase 1 \& 2,} \\ \text{Conv}_{1\times1}(\cdot), & \text{phase 3,} \end{cases} \quad (4)$$

where $f_l^F$ and $f_l^O$ denote the intermediate feature maps extracted from **Flash-VAED** and the original VAE decoder in block $l$, respectively, $\text{numel}(\cdot)$ is the normalization factor that represents the total element count, and $\sigma(\cdot)$ serves as a phase-dependent adaptation layer that aligns the features of Flash-VAED with the original feature space.

Based on this, we propose a three-phase dynamic distillation training strategy. In phase 1, we aim to capture global structural information by directly aligning the features of the deep layers of Flash-VAED with those of the original

VAE decoder. The complete loss function is formulated as

$$L = \alpha_1 L_1 + \alpha_2 L_{\text{lpips}} + \alpha_3 L_{\text{distill}} + \alpha_4 L_{\text{ssim}}. \quad (5)$$

Then, phase 2 focuses on enhancing pre-pruned channels by incorporating $L_{ce}$ from Equation 3 into the overall objective to maximize the expressivity of retained channels. The complete loss function for this phase is represented as

$$L = \alpha_1 L_1 + \alpha_2 L_{\text{lpips}} + \alpha_3 L_{\text{distill}} + \alpha_4 L_{\text{ssim}} + \alpha_5 L_{\text{ce}}. \quad (6)$$

Finally, phase 3 targets fine-grained recovery in pruned shallow layers. To address channel number misalignment caused by pruning, we instantiate $\sigma$ as a $1 \times 1$ convolution. Notably, we initialize this projection layer using $\mathbf{W}$ derived in Section 3.1 to significantly accelerate convergence. The loss function for this phase is the same as Equation 5.

## 4. Experiments

In this section, we first describe the experiment setup, including the implementation details of our Flash-VAED family, training and benchmark datasets, baselines, and evaluation protocols in Section 4.1. Then, we present our main results regarding both quality and efficiency in Section 4.2. Finally, we conduct ablation studies to justify the design choices for critical components and parameters in Section 4.3.

### 4.1. Experimental Setup

**Flash-VAED Family.** To demonstrate the versatility and robustness of our approach, we instantiate the Flash-VAED family by applying our optimization methods to the decoders of two representative state-of-the-art (SOTA) video VAEs: Wan (Wan et al., 2025) and LTX-Video (HaCohen et al., 2024). The rationale for this selection is twofold. First, we choose Wan (Wan et al., 2025) because its VAE decoder is widely recognized for delivering the highest quality among mainstream video generation models, serving as a benchmark for high-fidelity performance. Second, LTX-Video (HaCohen et al., 2024), which operates at an extreme spatial-temporal compression ratio of 1:192, demonstrates the superior efficiency and effectiveness of our method even under ultra-high compression regimes.

**Datasets.** To ensure robustness for both reconstruction and generation tasks, we curate a hybrid training dataset that comprises 10K samples, equally drawn from two distinct sources. The first half consists of 5K video-latent pairs obtained through video reconstruction, where videos sampled from the VidGen (Tan et al., 2024) dataset and latents are extracted using the original VAE encoder. The second half comprises 5K latent-video pairs generated via a pipeline where diverse text prompts generated by GPT-4o are fed into the video generation model (Wan et al., 2025; HaCohen et al., 2024) to collect pairs of final denoised latents and

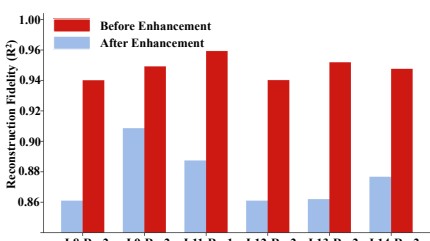
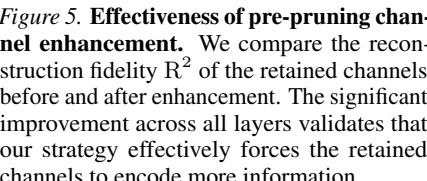

*Figure 5.* **Effectiveness of pre-pruning channel enhancement.** We compare the reconstruction fidelity $R^2$ of the retained channels before and after enhancement. The significant improvement across all layers validates that our strategy effectively forces the retained channels to encode more information.

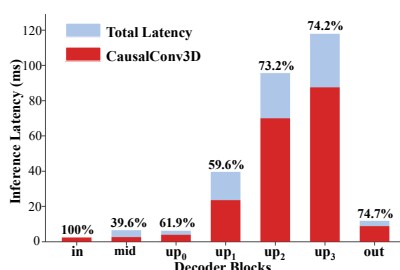

*Figure 6.* **Block-wise latency breakdown.** The CausalConv3D operator occupies over **60%** of the total latency in most blocks, and its inference time grows rapidly with resolution. These observations jointly lead us to define it as the **dominant operator**.

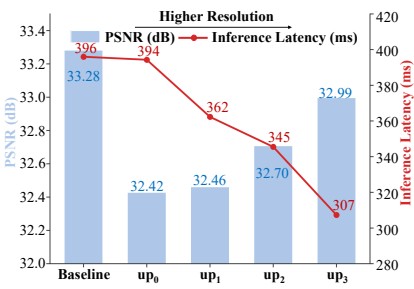

*Figure 7.* **Analysis of 2D convolution substitution.** Moving towards higher resolution layers, replacing CausalConv3D with 2D convolutions consistently reduces the latency while delivering less quality loss.

generated videos. For quantitative evaluation, following prior work (Zou et al., 2026), we utilize the UCF-101 test set (Soomro et al., 2012) as our reconstruction benchmark.

**Baselines.** We compare our Flash-VAED against the following baselines in terms of both quality and efficiency:

- **Turbo-VAED** (Zou et al., 2026): The current SOTA framework for accelerating VAE decoders specifically for video generation tasks.
- **LightVAE** (Contributors, 2025): An open-source solution in the Hugging Face LightX2V series, known for its strong practical adoption, with over 7,000 monthly downloads.

**Metrics.** To comprehensively evaluate our method, we first assess the reconstruction quality using four standard metrics: PSNR for pixel-level fidelity, SSIM for structural preservation, LPIPS for perceptual realism aligned with human vision, and warping error (Lai et al., 2018b) for temporal consistency. To rigorously verify Flash-VAED's capability on compute-constrained devices, we benchmark inference speed (FPS) on both a consumer-grade GPU (NVIDIA RTX 5090D) and an edge device (NVIDIA Jetson AGX Orin 64G). For the generation task, we measure end-to-end latency to evaluate efficiency. The quality evaluation is conducted using VBench 2.0 (Zheng et al., 2025), the most authoritative and comprehensive benchmark for assessing models across 18 distinct dimensions, ensuring that our acceleration maintains high-quality generation standards with holistic evaluation.

**Implementation Details.** Flash-VAED is trained on video sequences with a resolution of $480\times832$ spanning 81 frames. The training process is conducted on four NVIDIA A5880 GPUs and requires approximately 300 GPU-hours. We employ the AdamW optimizer with a learning rate of $1 \times 10^{-4}$ and a weight decay of $1 \times 10^{-4}$. The loss coefficients are set as $\alpha_1 = 10$, $\alpha_2 = 2$, $\alpha_3 = 1$, and $\alpha_4 = 5$ during the first and third training stages, while in the second stage, we

maintain these values and additionally set $\alpha_5 = 1$. In terms of specific architectural adaptations, for the Wan 2.1 VAE decoder (Wan et al., 2025), we replace the blocks mid, $up_0$, and $up_1$ with depthwise separable convolutions, and switch $up_2$ and $up_3$ to 2D convolutions, applying $1/8$ channel pruning to the latter two blocks. Similarly, for the LTX-Video VAE decoder (HaCohen et al., 2024), we replace the blocks mid, $up_0$, and $up_1$ with depthwise separable convolutions, keep $up_2$ unchanged, replace $up_3$ with a 2D convolution, and apply $1/4$ channel pruning to both $up_2$ and $up_3$.

### 4.2. Main Results

**Reconstruction Results.** We present qualitative visual results in Figure 1 and report quantitative evaluations in Table 1 and Table 2. Table 1 compares reconstruction quality and inference speed, while Table 2 further evaluates temporal consistency using warping error (Lai et al., 2018b). For the Wan 2.1 VAE decoder (Wan et al., 2025), Flash-VAED achieves substantial speedups of 6.16× on the NVIDIA RTX 5090 D and 5.69× on the Jetson Orin, while retaining 93.1% of the original VAE decoder's quality. Notably, our reconstructed PSNR reaches 37.61 dB, surpassing even the original VAE decoders of other SOTA video generation models (Kong et al., 2024; HaCohen et al., 2024; Yang et al., 2025), which demonstrates the effectiveness of our method on VAE decoders with mainstream compression ratios. Compared to the baseline LightVAE-Wan2.1 (Contributors, 2025), Flash-VAED achieves an impressive 5 dB gain in PSNR at the same inference speed. For the LTX-Video VAE (HaCohen et al., 2024), Flash-VAED achieves 5.71× and 5.63× speedups on the NVIDIA RTX 5090 D and Jetson Orin, respectively, while preserving 96.9% of the original quality. This further validates the efficacy of our method on VAE decoders with high compression ratios. Compared with Turbo-VAED (Zou et al., 2026), Flash-VAED outperforms in both speed and quality. Specifically,

*Table 1.* **Quantitative comparison of video reconstruction results with SOTA methods.** We compare Flash-VAED against the original VAE decoders and competitive baselines on both a consumer-grade GPU (RTX 5090D) and an edge device (Jetson Orin), where $d_T, d_H$, and $d_W$ denote the compression ratios for time, height, and width, respectively, $\uparrow$ indicates higher is better, while $\downarrow$ indicates lower is better. The best results among the methods are highlighted in **bold**. Flash-VAED outperforms all the baselines in both speed and quality.

| Model | $(d_T, d_H, d_W)$ | FPS $\uparrow$ | | Reconstruction Quality | | |
| --- | --- | --- | --- | --- | --- | --- |
| | | RTX 5090D | Jetson Orin | PSNR $\uparrow$ | SSIM $\uparrow$ | LPIPS $\downarrow$ |
| Wan 2.1 (Wan et al., 2025) | $(4, 8, 8)$ | 19.27 | 0.65 | 40.40 | 0.9733 | 0.0190 |
| LightVAE-Wan 2.1 (Contributors, 2025) | $(4, 8, 8)$ | 118.60 | **3.70** | 32.61 | 0.9416 | 0.0892 |
| **Flash-VAED-Wan 2.1 (Ours)** | $(4, 8, 8)$ | **118.77** | **3.70** | **37.61** | **0.9614** | **0.0285** |
| LTX-Video (HaCohen et al., 2024) | $(8, 32, 32)$ | 204.55 | 4.75 | 33.28 | 0.9253 | 0.0497 |
| Turbo-VAED-LTX (Zou et al., 2026) | $(8, 32, 32)$ | 623.08 | 23.24 | 31.52 | 0.9275 | 0.0555 |
| **Flash-VAED-LTX (Ours)** | $(8, 32, 32)$ | **1167.99** | **26.74** | **32.24** | **0.9293** | **0.0551** |

*Table 2.* Quantitative comparison of temporal consistency with SOTA methods. We report warping error to measure temporal coherence, where $\downarrow$ indicates lower is better. The best results among the methods are highlighted in bold.

| Model | $E_{\mathrm{warp}} \downarrow$ |
| --- | --- |
| Wan 2.1 (Wan et al., 2025) | 0.017783 |
| LightVAE-Wan 2.1 (Contributors, 2025) | 0.026640 |
| **Flash-VAED-Wan 2.1** | **0.018662** |
| LTX-Video (HaCohen et al., 2024) | 0.022071 |
| Turbo-VAED-LTX (Zou et al., 2026) | 0.026261 |
| **Flash-VAED-LTX** | **0.024267** |

Flash-VAED is nearly $2\times$ faster than Turbo-VAED on the NVIDIA RTX 5090 D and even faster on the Jetson Orin, without specific edge optimizations. For quality comparison, our proposed Flash-VAED surpasses Turbo-VAED across all PSNR, SSIM, and LPIPS metrics.

Beyond frame-wise reconstruction quality and inference efficiency, we further evaluate the temporal coherence of reconstructed videos by evaluating the warping error (Lai et al., 2018b). Given a video sequence, let $x_t$ denote the ground-truth frame at time $t$, and $\hat{x}_t$ denote the reconstructed frame. We first estimate the optical flow between adjacent ground-truth frames using a pre-trained RAFT model:

$$F_t = \mathrm{RAFT}(x_t, x_{t-1}), \qquad (7)$$

where $F_t$ represents the motion field from frame $t-1$ to frame $t$. The warping error is computed by warping the previous reconstructed frame $\hat{x}_{t-1}$ to the current time step using $F_t$, and measuring its discrepancy from the current reconstructed frame $\hat{x}_t$. Formally, the warping error $E_{\mathrm{warp}}$ is defined as:

$$E_{\mathrm{warp}} = \frac{1}{T-1} \sum_{t=2}^{T} \|\hat{x}_t - \mathrm{Warp}(\hat{x}_{t-1}, F_t)\|_1. \quad (8)$$

A lower warping error indicates better temporal consistency, since temporally coherent reconstructions should remain well aligned after motion compensation. As shown in Table 2, Flash-VAED achieves lower warping error than the corresponding baselines on both the Wan 2.1 (Wan et al., 2025) and LTX-Video (HaCohen et al., 2024) backbones. These results demonstrate that Flash-VAED delivers significant acceleration while incurring only minimal temporal-consistency degradation.

This superiority arises from (i) a lightweight model design that facilitates fast inference, and (ii) an efficient training strategy that enables Flash-VAED to retain the capabilities of the original VAE decoders.

**Generation Results.** To demonstrate the versatility of our method, we integrate Flash-VAED into two representative accelerated generation frameworks: Self Forcing-Wan 1.3B, an autoregressive acceleration framework (Huang et al., 2026), and FastVideo-Wan 1.3B, a few-step distillation acceleration framework (Zhang et al., 2025). The video generation results are qualitatively visualized in Figure 8 for Self Forcing-Wan 1.3B (left) and FastVideo-Wan 1.3B (right), with the corresponding quantitative results evaluated on VBench 2.0 reported in Figure 9 on the left and right sides, respectively. For the Self Forcing pipeline (Huang et al., 2026), Flash-VAED achieves a 27% speedup in end-to-end generation latency, while the performance curve in Figure 9a closely tracks that of the original VAE decoder across all dimensions in VBench-2.0 (Zheng et al., 2025). In contrast, the baseline LightVAE (Contributors, 2025), although comparable (slightly slower) in speed, demonstrates noticeable performance drops in several dimensions. The advantages of Flash-VAED are even more pronounced in the FastVideo pipeline (Zhang et al., 2025), where it achieves a 36% speedup while maintaining a closely aligned performance curve with the original VAE decoder. Conversely, the performance of the LightVAE-Wan 2.1 baseline degrades significantly, resulting in the decoding of latent information into meaningless noise. This disparity comes from the mismatch in latent distribution, as illustrated in Figure 10, which visualizes the latent distribution of the original VAE, Flash-VAED, and the baseline. It is evident that the latent distribution of Flash-VAED (red) perfectly overlaps with that of the original VAE decoder (black), ensuring seam-

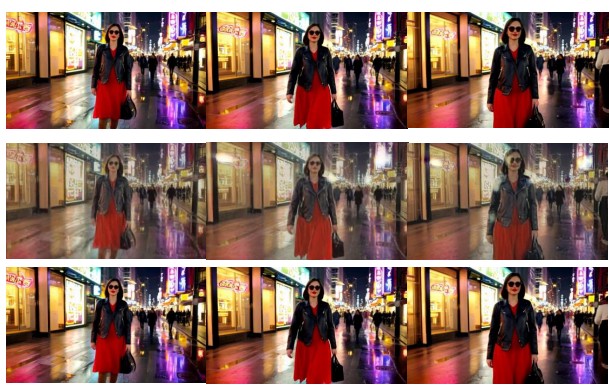

**Prompt example:** A stylish woman strolls down a bustling Tokyo street, the warm glow of neon lights and animated city signs casting vibrant reflections. She wears a sleek black leather jacket paired with a flowing red dress and black boots, her black purse slung over her shoulder.

**(a) Self Foring-Wan 1.3B**

**Prompt example:** A fluffy golden retriever sits calmly in a sunlit backyard, surrounded by vibrant green grass and colorful flower beds, its eyes alert and ears perked. Suddenly, with a burst of energy, it leaps up and begins to run in joyful circles, tail wagging furiously.

**(b) FastVideo-Wan 1.3B**

Top: Wan 2.1 VAE  Middle: LightVAE-Wan 2.1  Bottom: Flash-VAED-Wan 2.1

*Figure 8.* **Visual comparison of video generation results.** Our Flash-VAED (bottom) matches closely with the original Wan 2.1 VAE (top) in fidelity, details and textures, whereas the LightVAE (middle) exhibits severe artifacts that results in invalid video content. The full prompts and additional details are provided in Appendix A.4.

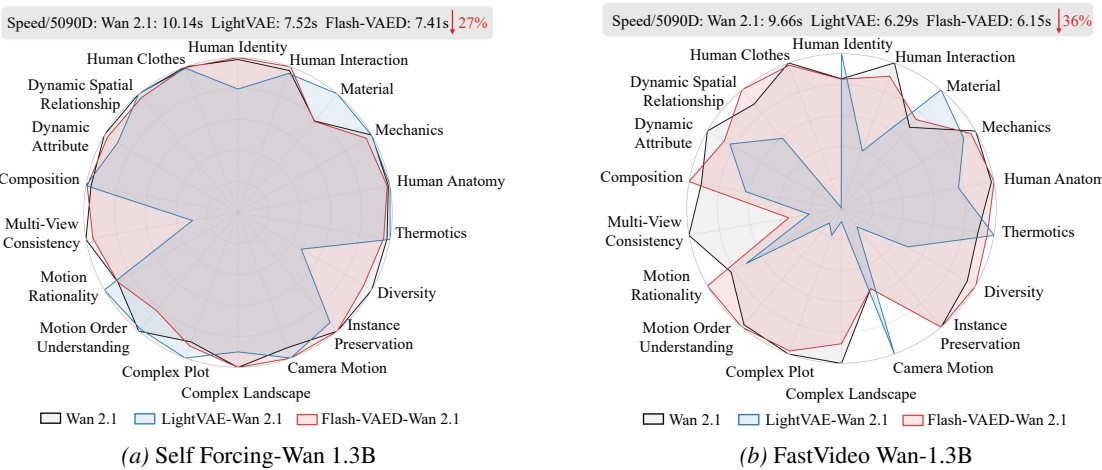

*(a)* Self Forcing-Wan 1.3B      *(b)* FastVideo Wan-1.3B

*Figure 9.* Generation performance of Flash-VAED embedded on the VBench-2.0. We employ Flash-VAED to replace the VAE decoders of: (a) the Self Forcing-Wan 1.3B, and (b) the FastVideo-Wan 1.3B, whose overall generation quality is evaluated on VBench-2.0.

---

less compatibility with the generation pipeline, while Light-VAE (blue) (Contributors, 2025) suffers from a significant distributional deviation.

### 4.3. Ablation Study

**Ablation on Pruning Methods.** To validate our proposed independence-aware channel pruning strategy, we compare it with two classic pruning methods: random pruning and L1-norm based channel pruning. For the L1-norm based channel pruning, the importance of each channel is estimated by the L1-norm of its corresponding channel weights. As shown in Figure 11, our method demonstrates a significantly higher starting point, faster performance growth, and

superior outcomes. These improvements verify that our independence-aware channel selection effectively preserves the most informative channels, laying a solid foundation for the retained-channel enhancement in the second stage, while ensuring that the retained channels effectively inherit the capabilities from the original full channel set.

**Ablation on Pruning Ratios.** Taking the LTX-Video VAE decoder (HaCohen et al., 2024) as a representative example, we investigate the speed-quality trade-off across different pruning ratios $(1, 1/2, 1/4, 1/8)$, as presented in Table 3. As the pruning ratio decreases from 1 to $1/8$, the inference speed steadily increases, with a degradation in performance. We select a 1/4 ratio for the LTX-Video VAE decoder (HaCo-

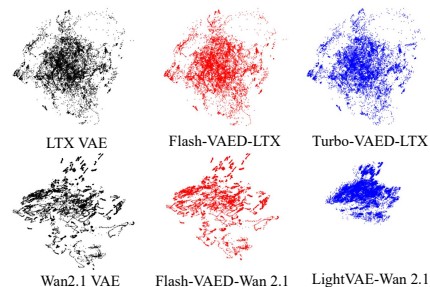

*Figure 10.* Visualization of samples from the latent distribution.

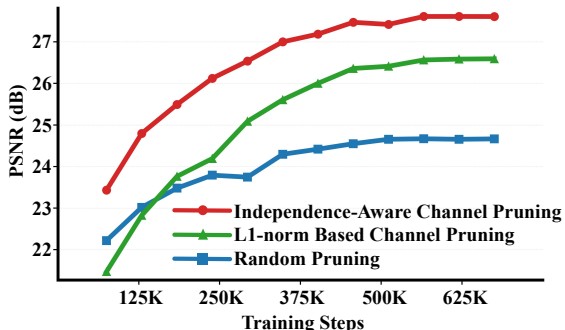

*Figure 11.* Ablation results of using different pruning methods.

*Table 3.* Ablation results of using different pruning ratios.

| Prune Ratio | FPS/5090D ↑ | PSNR ↑ | SSIM ↑ | LPIPS ↓ |
|---|---|---|---|---|
| 1 | 204.55 | 33.2883 | 0.9253 | 0.0497 |
| 1/2 | 625.00 | 32.5421 | 0.9307 | 0.0521 |
| 1/4 | 1167.99 | 32.2400 | 0.9293 | 0.0551 |
| 1/8 | 1907.23 | 31.0786 | 0.9162 | 0.0804 |

*Table 4.* Ablation results of different projected layer initialization.

| Project Layer | PSNR ↑ | SSIM ↑ | LPIPS ↓ |
|---|---|---|---|
| Random Initialization | 30.3061 | 0.9013 | 0.0978 |
| **Ours** | **32.2400** | **0.9293** | **0.0551** |

*Table 5.* Ablation results of using different distillation methods.

| Training Methods | PSNR ↑ | SSIM ↑ | LPIPS ↓ |
|---|---|---|---|
| Without Distillation | 30.7957 | 0.9125 | 0.0831 |
| Deep-only Distillation | 30.8065 | 0.9126 | 0.0830 |
| Shallow-only Distillation | 31.2386 | 0.9152 | 0.0722 |
| **Ours** | **32.3768** | **0.9317** | **0.0566** |

uitous presence and high inference cost of CausalConv3D operations. To address these challenges, we propose two targeted strategies: independence-aware channel pruning and stage-wise dominant operator optimization. Building on these optimizations, we introduce the Flash-VAED family. Furthermore, to facilitate the efficient transfer of capabilities from the original VAE decoders to Flash-VAED, we develop a three-phase dynamic distillation training pipeline. Extensive experiments demonstrate that Flash-VAED significantly outperforms baselines, achieving the fastest decoding speed with minimal loss of fidelity to the original VAE decoders.

hen et al., 2024) as it achieves the best trade-off: compared with a $1/2$ ratio, the $1/4$ ration provides a $2\times$ speedup with only a 0.93% drop in quality, while the $1/8$ ratio incurs a more substantial 3.7% quality loss. Similarly, for the Wan 2.1 VAE decoder (Wan et al., 2025), we identify the $1/8$ ratio as the optimal configuration that effectively balances speed and quality.

**Ablation on Training Strategies.** To validate the rationale behind our three-phase dynamic distillation pipeline, we compare our method against settings without distillation and those applying distillation exclusively to shallow or deep layers. As shown in Table 5, our method maximizes the transfer of capabilities from the original VAE decoders to Flash-VAED, achieving 96.9% of the original quality. Moreover, we evaluate the impact of initializing the projection layer using the $W$ matrix derived from the second training stage during the third training stage, comparing it against random initialization. As indicated in Table 4, our initialization method leads to approximately a 6.25% improvement in reconstruction quality.

## 5. Conclusions

In this paper, we focus on accelerating VAE decoders for video generation while ensuring strict alignment with the original latent distribution. We first experimentally identify two primary factors that contribute to the latency bottleneck of VAE decoders: extreme channel redundancy and the ubiq-

## Acknowledgement

This work was supported by the Hong Kong Research Grants Council under the Areas of Excellence scheme grant AoE/E-601/22-R and NSFC/RGC Collaborative Research Scheme grant CRS_HKUST603/22.

## Impact Statement

This paper presents work whose goal is to advance the field of Machine Learning. There are many potential societal consequences of our work, none of which we feel must be specifically highlighted here.

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

# A. Implementation Details

## A.1. Mechanism of Depthwise Separable Convolutions

Standard 3D convolutions effectively model spatiotemporal correlations (across time, height, and width) and cross-channel correlations. While effective, this approach incurs high computational costs. To address this, depthwise separable 3D convolution decomposes this operation into two efficient stages: spatiotemporal filtering and channel mixing. We use $T, H,$ and $W$ to denote the number of frames, the height, and the width of the videos, respectively, while $C_{\text{in}}$ and $C_{\text{out}}$ represent the input and output channels of the convolution layers, respectively.

**Standard 3D Convolution.** Given an input video tensor of size $T \times H \times W$, a standard 3D convolution with a cubic kernel size of $N \times N \times N$ incurs a computational cost (measured in FLOPs) of:

$$\text{Cost}_{\text{std}} = T \cdot H \cdot W \cdot C_{\text{in}} \cdot C_{\text{out}} \cdot N^3. \tag{9}$$

**Depthwise Separable 3D Convolution.** The proposed mechanism splits the computation into two steps:

1. **Depthwise Convolution:** We first apply spatiotemporal filters of size $N \times N \times N$ to each input channel independently. This step captures both motion and spatial patterns while maintaining channel independence, with a computational cost (in FLOPs) of:

$$\text{Cost}_{\text{dw}} = T \cdot H \cdot W \cdot C_{\text{in}} \cdot N^3. \tag{10}$$

2. **Pointwise Convolution:** We then apply a $1 \times 1 \times 1$ convolution to linearly combine the filtered features across channels, with a computational cost (in FLOPs) of:

$$\text{Cost}_{\text{pw}} = T \cdot H \cdot W \cdot C_{\text{in}} \cdot C_{\text{out}}. \tag{11}$$

**Efficiency Analysis.** The total cost of the depthwise separable 3D convolution, denoted by $\text{Cost}_{\text{sep}}$, is the sum of $\text{Cost}_{\text{dw}}$ and $\text{Cost}_{\text{pw}}$, i.e., $\text{Cost}_{\text{sep}} = \text{Cost}_{\text{dw}} + \text{Cost}_{\text{pw}}$. The reduction ratio of the computational cost relative to standard 3D convolution is derived as:

$$\frac{\text{Cost}_{\text{sep}}}{\text{Cost}_{\text{std}}} = \frac{T \cdot H \cdot W \cdot C_{\text{in}}(N^3 + C_{\text{out}})}{T \cdot H \cdot W \cdot C_{\text{in}} \cdot C_{\text{out}} \cdot N^3} = \frac{1}{C_{\text{out}}} + \frac{1}{N^3}. \tag{12}$$

For a typical kernel size where $N = 3$, the term $\frac{1}{N^3}$ is equal to $\frac{1}{27}$. As $C_{\text{out}}$ increases, $\frac{1}{C_{\text{out}}}$ becomes negligible. Consequently, this design reduces the computational load and parameter count by approximately 27 times compared to standard 3D convolutions.

## A.2. Detailed Analysis of the Effectiveness of Shortcut Injection

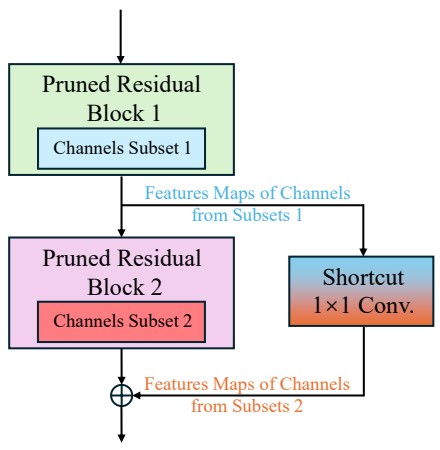

*Figure 12.* **Effectiveness of shortcut injection.**

In this section, we detail why retaining different indices between residual blocks destroys the continuity of the original model. We also explain how our pruning strategy and $1 \times 1$ convolutional shortcut injection effectively address this issue.

A standard residual block consists of two parallel pathways: the main path, which comprises stacked convolutional layers that transform the features, and the shortcut path, which directly links the input to the output to prevent gradient vanishing.

Typically, an identity mapping is employed for the shortcut path. However, when the retained channel indices of two consecutive blocks are different, using an identity shortcut become problematic. Specifically, it results in the addition of feature maps from Channel Subset 1 (the output of Residual Block 1) to feature maps from Channel Subset 2 (the output of Residual Block 2), thereby destructing the continuity of the original model. To address this, our pruning strategy ensures that the feature maps from the retained channels of Residual Block 1 can linearly represent the feature map from any channel within the same block. Consequently, we

can establish a mapping from these retained channels to the specific channels in Residual Block 1 that correspond to the retained indices of Residual Block 2. This mapping matrix, denoted by $\mathbf{W}$, is computed using the least squares method and is then used to initialize an injected $1 \times 1$ convolutional shortcut. This approach aligns the channel features from the shortcut pathway with those from the main path of the Pruned Residual Block 2, allowing for the effective summation of outputs and thus restoring connectivity within the model.

As for the main path, although no projection layer is injected, it does not suffer from topological discontinuity. Since the feature maps output from Residual Block 1 can linearly reconstruct the feature map from any channel, the subsequent convolutional layers in Residual Block 2 can readily learn the mapping from the input to the desired feature representation.

### A.3. Gradient Mask for Training Stage 2

In the second training stage, i.e., pre-pruning retained channel enhancement, we employ a gradient mask to prevent all channels from converging towards the retained channels. Specifically, for the to-be-pruned channels, we suspend weight updates, preserving their original information while compelling the retained channels to fit the full-channel representation.

### A.4. Additional Details for the Visual Comparison of Video Generation Results

Due to space constraints in the main text, we present only partial prompts. Here, we provide the complete prompts used for visual comparisons.

For the visual comparison of Self Forcing, the full prompt employed is: *A stylish woman strolls down a bustling Tokyo street, the warm glow of neon lights and animated city signs casting vibrant reflections. She wears a sleek black leather jacket paired with a flowing red dress and black boots, her black purse slung over her shoulder. Sunglasses perched on her nose and a bold red lipstick add to her confident, casual demeanor. The street is damp and reflective, creating a mirror-like effect that enhances the colorful lights and shadows. Pedestrians move about, adding to the lively atmosphere. The scene is captured in a dynamic medium shot with the woman walking slightly to one side, highlighting her graceful strides.*

For FastVideo, the full prompt employed is: *A fluffy golden retriever sits calmly in a sunlit backyard, surrounded by vibrant green grass and colorful flower beds, its eyes alert and ears perked. Suddenly, with a burst of energy, it leaps up and begins to run in joyful circles, tail wagging furiously. The camera captures its playful antics, zooming in on its gleeful expression and the way its fur catches the sunlight. As it races around, the flowers sway gently in the breeze, and the sound of its paws thumping on the ground adds to the lively atmosphere. Finally, it pauses, panting happily, before flopping down on the grass, basking in the warmth of the day.*

Additionally, for the FastVideo visual comparison, we replace the VAE decoder in the Fastvideo pipeline with the original VAE decoder from Wan 2.1 to ensure fairness. Note that Fastvideo recommends an inference resolution of $448 \times 832$, and its VAE decoder is specifically fine-tuned for this resolution. In contrast, both our VAE decoder and the Wan 2.1 VAE decoder are trained at a resolution of $480 \times 832$. Therefore, to maintain a fair comparison at the same resolution, we use the original Wan 2.1 VAE decoder. However, it is important to note that this replacement is not applied in quantitative evaluations.

## B. Supplementary Experimental Results

### B.1. Generalization to High-resolution and Complex Reconstruction Benchmark

To further evaluate the generalization capability of Flash-VAED, we conduct challenging experiments on the DAVIS 2019 Challenge test set (Caelles et al., 2019) at 720p resolution. Compared with UCF-101 (Soomro et al., 2012), DAVIS 2019 features highly diverse, complex in-the-wild videos characterized by fast object motion, heavy occlusions, dynamic camera movement, and fine-grained visual details. These characteristics make it a desired benchmark for assessing the robustness of a accelerated VAE decoder in high-resolution and temporally complex scenarios.

We present qualitative and quantitative results on DAVIS 2019 in Figure 13 -Figure 14, and Table 6 respectively. Flash-VAED consistently delivers superior reconstruction quality on both the Wan 2.1 (Wan et al., 2025) and LTX-Video (HaCohen et al., 2024) backbones, significantly outperforming baselines across all metrics. These results demonstrate that Flash-VAED generalizes effectively to high-resolution videos with complex motion and rich details, despite being trained only at 480p resolution. We attribute this zero-shot spatial generalization to the fully convolutional nature of Flash-VAED. Specifically, the decoder relies on local receptive fields to learn translation-equivariant mappings from latent representations to the pixel

domain, and is hence agnostic to fixed input resolutions. Consequently, the learned convolutional kernels seamlessly scale to larger latent grids during inference. Moreover, since low-level visual statistics, such as textures, edges, and local structures, are largely invariant across spatial resolutions, the local reconstruction priors trained from 480p remain effective when transferred to 720p evaluation.

*Table 6.* Reconstruction quality comparison on the DAVIS 2019 Challenge test set at 720p resolution, where ↑ indicates higher is better and ↓ indicates lower is better. The best results are highlighted in bold.

| Model | PSNR ↑ | SSIM ↑ | LPIPS ↓ |
|---|---|---|---|
| Wan 2.1 (Wan et al., 2025) | 34.13 | 0.9079 | 0.0487 |
| LightVAE-Wan 2.1 (Contributors, 2025) | 27.92 | 0.8397 | 0.2069 |
| **Flash-VAED-Wan 2.1** | **32.03** | **0.8825** | 0.0769 |
| LTX-Video (HaCohen et al., 2024) | 29.36 | 0.8134 | 0.1555 |
| Turbo-VAED-LTX (Zou et al., 2026) | 26.82 | 0.7805 | 0.1783 |
| **Flash-VAED-LTX** | **28.26** | **0.8031** | **0.1214** |

### B.2. Additional Efficiency Analysis

*Table 7.* Additional efficiency analysis on the Wan 2.1 VAE decoder. We report the parameter count and peak VRAM usage of different lightweight variants and methods.

| Model | Number of Parameters ↓ | Peak VRAM ↓ |
|---|---|---|
| Wan 2.1 (Wan et al., 2025) | 73.30 M | 7.11 GB |
| Wan 2.1 + Channel Pruning | 65.20 M | 2.31 GB |
| Wan 2.1 + Operator Optimization | 5.63 M | 3.51 GB |
| LightVAE-Wan 2.1 (Contributors, 2025) | 4.62 M | 1.17 GB |
| Flash-VAED-Wan 2.1 | 5.86 M | 2.00 GB |

Flash-VAED aims to improve the overall efficiency of VAE decoding, rather than merely increasing inference speed. To rigorously evaluate this, we take the Wan 2.1 VAE decoder as a testbed and compare Flash-VAED with representative baselines in terms of parameter count and peak GPU memory usage, measured by video random-access memory (VRAM). We also ablate the intermediate variants at each lightweighting operation to isolate the efficiency gains contributed by individual components.

As shown in Table 7, both Flash-VAED and LightVAE compress the original decoder to less than $10\%$ of its initial parameter count and reduce the peak VRAM to 2.00GB or lower. However, as evidenced by our main reconstruction results in Section 4.2, LightVAE suffers from severe quality degradation under such aggressive compression. In contrast, Flash-VAED achieves an extreme compact size while almost preserving the reconstruction quality of the original VAE decoder, thereby establishing a superior efficiency-fidelity trade-off.

The intermediate variants further illustrate the efficiency contribution of each lightweighting stage. The operator-optimization variant, denoted as *Wan 2.1 + Operator Optimization*, significantly decreases both parameter count and computational overhead. Conversely, the channel-pruning variant, denoted as *Wan 2.1 + Channel Pruning*, is specifically tailored for peak VRAM reduction, driving a drastic reduction in peak VRAM from 7.11 GB to 2.31 GB.

### B.3. Plug-and-Play Generalization to New Generation Pipelines

To further validate the plug-and-play capability and latent-space alignment of Flash-VAED, we extend its deployment to a pipeline that is distinct from the evaluation settings of the main body. Specifically, we replace the original Wan 2.1 VAE decoder (Wan et al., 2025) with Flash-VAED in Causal Forcing (Zhu et al., 2026), which is a recent representative method for autoregressive few-step distillation. We evaluate the generated videos using VBench 2.0 (Zheng et al., 2025) and report randomly selected metrics.

As shown in Table 8, Flash-VAED-Wan 2.1 closely matches the original Wan 2.1 decoder (Wan et al., 2025) across most of the VBench 2.0 dimensions. In contrast, LightVAE-Wan 2.1 (Contributors, 2025) exhibits substantial performance degradation. These results validate that Flash-VAED possesses robust generalization and plug-and-play capabilities, seamlessly aligning with the original latent space even when integrated into entirely new generation pipelines.

*Table 8.* Generation performance of Flash-VAED-integrated Causal Forcing on VBench 2.0. We replace the original Wan 2.1 VAE decoder with different accelerated decoders and report selected VBench 2.0 dimensions, including Dynamic Spatial Relationship (DSR), Human Clothes (HC), Human Identity (HId), Human Interaction (HIn), Instance Preservation (IP), Motion Order Understanding (MOU), and Multi-View Consistency (MVC).

| Model | DSR ↑ | HC ↑ | HId ↑ | HIn ↑ | IR ↑ | MOU ↑ | MVC ↑ |
|---|---|---|---|---|---|---|---|
| Wan 2.1 (Wan et al., 2025) | 0.217 | 0.986 | 0.785 | 0.600 | 0.924 | 0.131 | 0.424 |
| LightVAE-Wan 2.1 (Contributors, 2025) | 0.198 | 0.953 | 0.527 | 0.570 | 0.819 | 0.121 | 0.072 |
| **Flash-VAED-Wan 2.1** | **0.237** | **0.986** | **0.741** | **0.590** | **0.918** | **0.141** | **0.318** |

## B.4. Discussion on Generalization to Additional Backbones

To further examine the generalization capability of Flash-VAED, we extend our analysis to alternative VAE decoder backbones, including Hunyuan Video (Kong et al., 2024) and CogVideoX (Yang et al., 2025) VAE decoders. Architectural analysis reveals that these decoders exhibit computational bottlenecks and structural characteristics similar to those of the Wan 2.1 (Wan et al., 2025) and LTX-Video (HaCohen et al., 2024) decoders. Specifically, they rely on massive channel counts and the frequent usage of CausalConv3D, which collectively dominate the computational cost. Additionally, their decoder architectures similarly consist of cascaded residual blocks, with temporal upsampling mainly concentrated in low-resolution layers. This architectural homogeneity provides a strong basis for transferring the efficiency optimization principles of Flash-VAED to broader video VAE decoders.

Our empirical analysis on these additional backbones further supports this transferability. By applying SVD to the feature maps of randomly sampled layers, we find that retaining only 12.5% and 11.72% of singular values is sufficient to explain 99% of the total variance on Hunyuan Video (Kong et al., 2024) and CogVideoX (Yang et al., 2025) VAE decoders, respectively. This observation is consistent with our low-rank assumption, suggesting the presence of substantial channel redundancy across different architectures. Moreover, implementing our operator optimization strategy, which replaces 3D convolutions with 3D depthwise separable convolutions in low-resolution layers and 2D convolutions in high-resolution layers, reduces the parameter count to approximately 10% of the original size and achieves around 35% inference speedup for both Hunyuan Video (Kong et al., 2024) and CogVideoX (Yang et al., 2025) VAE decoders. These results indicate that the design principles of Flash-VAED extend well beyond Wan 2.1 (Wan et al., 2025) and LTX-Video (HaCohen et al., 2024), offering a high generalization potential to a wider range of video VAE decoders.

## B.5. Discussion on Architectural Generalization

To further discuss the generalization of Flash-VAED across diverse VAE architectures, we analyze how its two main lightweighting components can be adapted to different decoder designs, including non-causal convolutional VAEs and transformer-based VAEs.

For the pruning strategy, Flash-VAED does not strictly rely on causal convolutional structures. In our framework, high-resolution layers are first converted to 2D convolutions, which are inherently non-causal, and channel pruning is then performed after this operator replacement. The effectiveness of pruning on both Wan (Wan et al., 2025) and LTX-Video (HaCohen et al., 2024) suggests that our channel selection criterion remains applicable to non-causal convolutional structures. For Transformer-based VAE decoders, the same principle can potentially be extended to other structurally redundant modules, such as Feed-Forward Network (FFN) hidden dimensions or embedding dimensions.

For operator optimization, the specific efficient operator should be selected according to the dominant computational bottleneck of the target architecture. For non-causal convolution-based VAE decoders, replacing standard convolutions with depthwise separable convolutions remains a practical way to reduce parameter count and computational cost. For Transformer-based VAE decoders, where convolutions are no longer the dominant operators, the focus can shift toward accelerating attention modules with efficient alternatives, such as linear attention (Huang et al., 2025a) or other efficient attention mechanisms.

Overall, computational cost remains a common challenge across many areas of artificial intelligence (Li et al., 2025a; Xue et al., 2026); therefore, model compression (Liu et al., 2025b) and acceleration will continue to be meaningful and important research directions. Although the specific operators may vary across different VAE architectures, the underlying design principle of Flash-VAED is broadly applicable: identifying heavy and redundant components and replacing or pruning them with efficient counterparts. This provides a promising direction for extending Flash-VAED to a wider range of VAE decoder architectures.

## B.6. Ablation on Resolution and Number of Frames

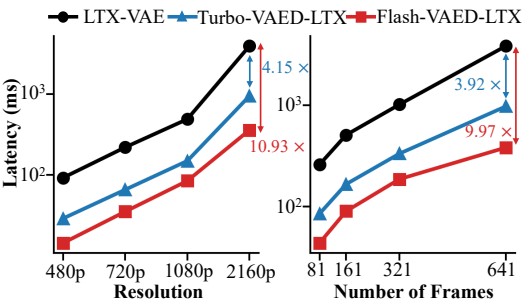

*Figure 15.* Ablation results of using different resolutions and numbers of frames.

To evaluate the performance of Flash-VAED in compute-intensive scenarios, we systematically increase the resolution and the number of frames in the decoded videos, and plot the latency variations in Figure 15. It is evident that as the resolution and number of frames increase, the latency of Flash-VAED scales more slowly compared with both the original VAE decoders and Turbo-VAED. Consequently, under the conditions of high resolution and large frame count, our method achieves a significant acceleration ratio, reaching approximately a $10\times$ speedup. The experiments confirm the superior scalability of Flash-VAED, and unleash its potential for high-compute scenarios.

## B.7. Latency Breakdown of Generation Pipelines

To further verify the latency bottleneck shift induced by the acceleration of the DiT, we conduct a component-wise latency breakdown of the full generation pipeline. We evaluate the original Wan 1.3B model (Wan et al., 2025) alongside the Self Forcing (Huang et al., 2026) and FastVideo (Zhang et al., 2025) optimized variants, as detailed in Table 9. The results indicate that text encoding contributes only a negligible portion of the total latency in both the baseline and accelerated pipelines. In contrast, the relative latency associated with the decoding stage becomes significantly more pronounced, with its percentage contribution increasing by approximately $15\times$ following the DiT acceleration.

*Table 9.* Latency breakdown of generation pipelines

| Model | Text Encoding | Denoising | Video Decoding | Decoding Percentage |
|---|---|---|---|---|
| Wan 1.3B (Wan et al., 2025) | 0.619 | 178.066 | 4.212 | 2.3% |
| Self Forcing-Wan 1.3B (Huang et al., 2026) | 0.598 | 6.254 | 3.172 | 31.6% |
| FastVideo-Wan 1.3B (Zhang et al., 2025) | 0.539 | 5.136 | 3.925 | 40.9% |

## B.8. Comparison of SOTA video VAE decoders

*Table 10.* To validate our choices of Wan (Wan et al., 2025) and LTX-Video (HaCohen et al., 2024) as original VAE decoders, we compare them against current SOTA video VAE decoders. Wan establishes a clear lead in both quality and speed among models with mainstream compression ratios, while LTX-Video dominates in terms of inference speed.

| Model | $(d_T, d_H, d_W)$ | FPS ↑ | Reconstruction Quality | | |
|---|---|---|---|---|---|
| | | RTX 5090D | PSNR ↑ | SSIM ↑ | LPIPS ↓ |
| Wan 2.1 (Wan et al., 2025) | $(4, 8, 8)$ | 19.27 | 40.40 | 0.9733 | 0.0190 |
| HunyuanVideo (Kong et al., 2024) | $(4, 8, 8)$ | 6.77 | 33.05 | 0.9392 | 0.0272 |
| CogVideoX (Yang et al., 2025) | $(4, 8, 8)$ | 4.99 | 32.81 | 0.9351 | 0.0302 |
| LTX-Video (HaCohen et al., 2024) | $(8, 32, 32)$ | 204.55 | 33.28 | 0.9253 | 0.0497 |

## B.9. Limitations and Failure Cases

Although Flash-VAED achieves substantial acceleration without compromising reconstruction quality in most scenarios, we may still observe its performance degradation in extreme cases. The qualitative comparisons in Figure 13 and Figure 14 show that Flash-VAED consistently outperforms the baselines in high-resolution and dynamic scenes. Nevertheless, videos containing a large number of rapidly moving tiny objects or densely appearing human faces remain challenging. Such cases usually involve dense high-frequency details and complex temporal variations, imposing stronger demands on the representational capacity of the decoder. Consequently, Flash-VAED may still exhibit motion blur, local distortions, or degraded facial details in these particularly challenging regions.

This limitation reflects the inherent trade-off between model efficiency and fine-grained reconstruction capability. Due to its

highly lightweight architecture, Flash-VAED has limited parameter to precisely reconstruct small, fast-moving structures or frequent appearance changes. As a result, minor artifacts, temporal flickering, or the loss of fine details may appear under stringent conditions.

Future work may tackle these challenging cases by introducing adaptive capacity allocation, motion-aware (Liu et al., 2025a) refinement modules, or specialized enhancement strategies in high-frequency regions. These directions could strengthen the representation of fine spatial details and rapid temporal dynamics while preserving the efficiency advantages of Flash-VAED.

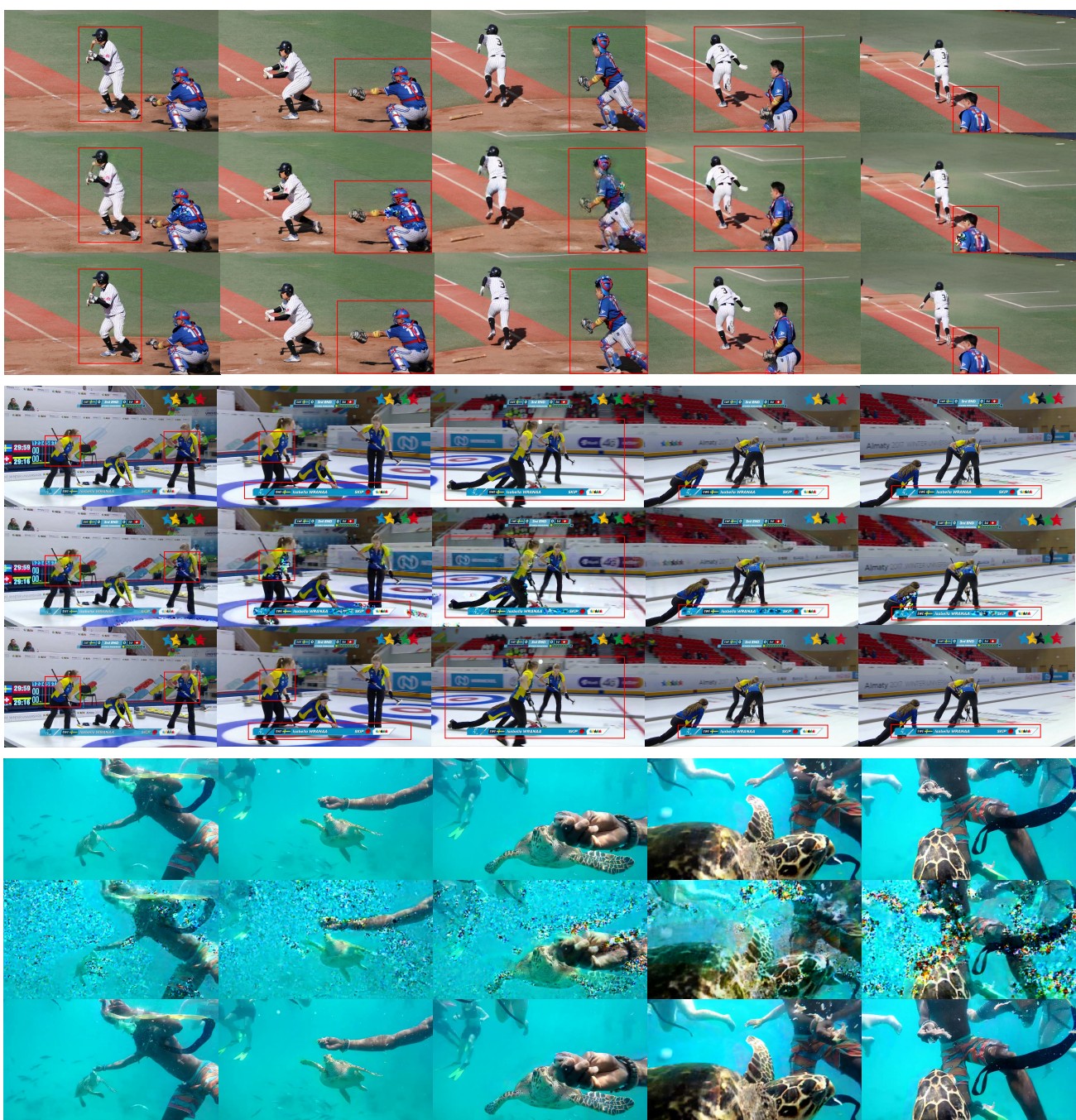

*Figure 13.* Qualitative comparison of video reconstruction results on the DAVIS 2019 Challenge test set for Wan 2.1-based VAE decoders. For each sample, the three rows from top to bottom correspond to the original Wan 2.1 decoder, LightVAE-Wan 2.1, and Flash-VAED-Wan 2.1, respectively. In high-resolution videos with fine-grained details and highly dynamic scenes, Flash-VAED preserves visual details and temporal structures better than LightVAE.

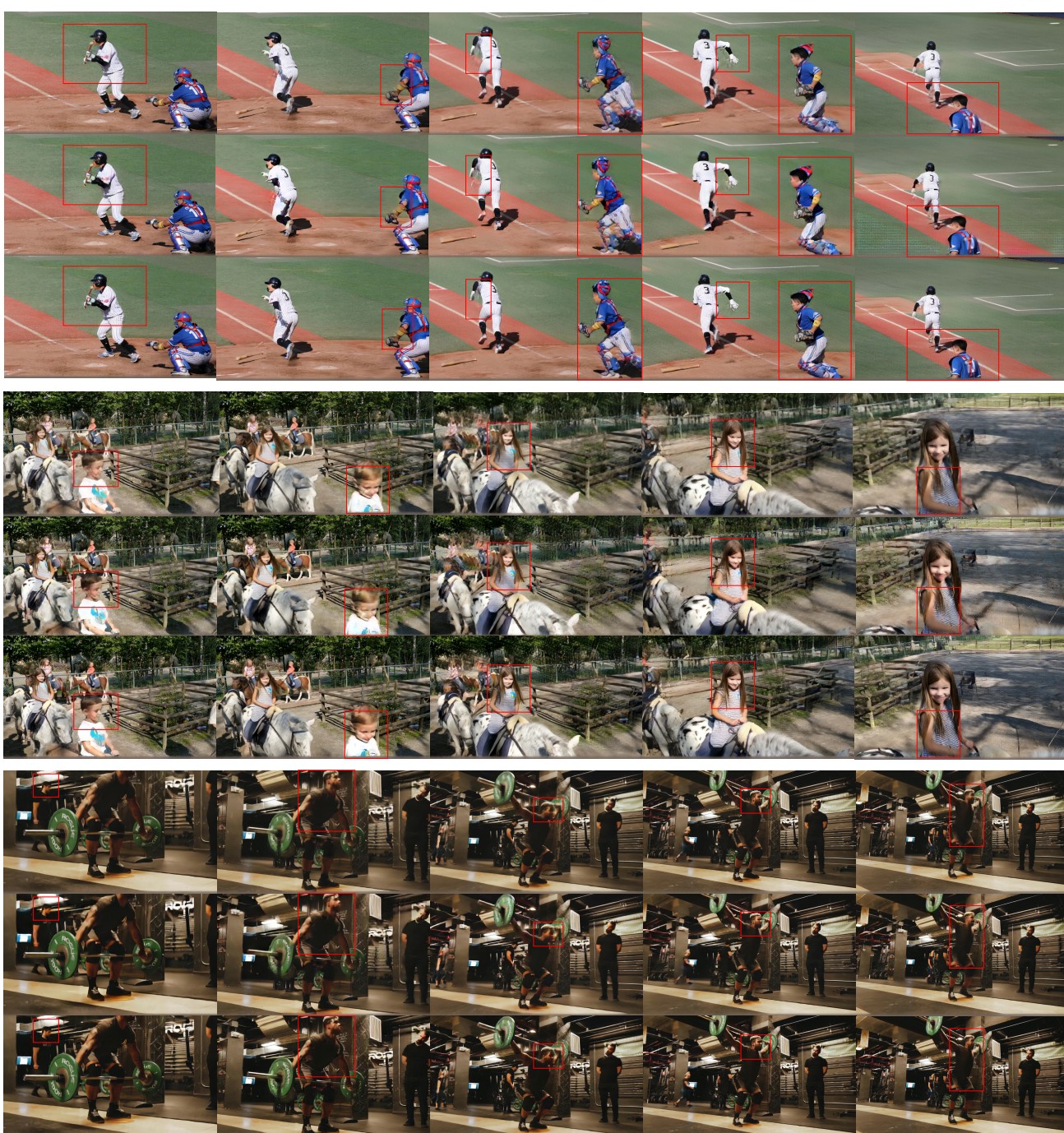

*Figure 14.* Qualitative comparison of video reconstruction results on the DAVIS 2019 Challenge test set for LTX-Video-based VAE decoders. For each sample, the three rows from top to bottom correspond to the original LTX-Video decoder, Turbo-VAED-LTX, and Flash-VAED-LTX, respectively. In high-resolution videos with fine-grained details and highly dynamic scenes, Flash-VAED achieves better reconstruction quality than Turbo-VAED.

