# OpenReview forum: "Flash-VAED: Plug-and-Play VAE Decoders for Efficient Video Generation"
_ICML.cc/2026/Conference — ICML 2026 regular_

### Official Review · Reviewer_mwsW · 2026-03-02

**Soundness:** 4
**Presentation:** 4
**Significance:** 3
**Originality:** 3
**Overall Recommendation:** 5
**Confidence:** 3

**Summary:**

This paper analyzes the inference bottlenecks inherent in current video VAE decoders and proposes two primary optimizations: (1) channel pruning and (2) replacing Causal 3D convolution with efficient operators. To implement these enhancements, the authors introduce a three-phase distillation to construct the Flash-VAED model family. Evaluations on Wan and LTX-Video VAEs demonstrate a 6x speedup in decoding latency with no noticeable loss in quality.

**Compliance With Llm Reviewing Policy:**

Affirmed.

**Final Justification:**

After reading other reviewer's comments and rebuttals, I maintain my positive assessment of the work and keep my positive assessment for this work.

**Key Questions For Authors:**

Please also refer to the weakness section.

1. What were the specific video configurations (resolution and frame count) used during the VBench-2.0 evaluation?
2. The Flash-VAED decoders were trained on 480x832 resolution video sequences. Since the baseline models (Wan 2.1/LTX-Video) frequently target 720p or 1080p outputs, can the authors provide evidence or discussion on how well these decoders generalize to resolutions higher than those seen during training?
3. Could you clarify which specific Wan 2.1 variants were utilized in the experiments to ensure full reproducibility?
4. The original Wan 2.1 decoder employs specific caching behaviors to enable the generation of high-resolution, long-form videos within memory constraints. Does Flash-VAED preserve these behaviors, or do the architectural changes (like 2D convolution substitution) interfere with existing memory-optimization strategies?

**Limitations:**

- I encourage the authors to include a "Failure Case Analysis" in the next revision to identify specific scenarios (e.g., extremely fine textures or complex temporal motions) where the 6x acceleration might lead to visible artifacts.
- Discussing the potential limitations regarding the "plug-and-play" nature when applied to VAEs with vastly different architectures (e.g., non-causal or transformer-based VAEs) would also add value.

**Strengths And Weaknesses:**

# Paper Strengths

1. The manuscript is exceptionally well-structured and easy to navigate. The inclusion of high-visualizations serves as an effective aid in elucidating the core technical particulars.
2. The motivation is both insightful and well-justified, shifting the focus to the VAE decoder as the emerging latency bottleneck in accelerated diffusion pipelines. The proposed methods are supported by sound technical rationale and reinforced by theoretical analysis.
3. The authors provide an exhaustive evaluation across two representative VAE decoders: Wan 2.1 (high fidelity) and LTX-Video (high compression). The inclusion of benchmarks on edge devices (Jetson Orin) significantly strengthens the paper’s claims regarding practical deployment. The experimental analysis is thorough and provides meaningful insights into the speed-quality trade-offs.

# Paper Weakness

While the evaluation is extensive, the reliance on the UCF-101 test set for quantitative reconstruction metrics is a minor weakness. Given its relatively low resolution and narrow focus on sports-related human actions, it may not fully capture the decoder’s performance on the diverse, high-resolution content that modern models like Wan 2.1 are designed to generate.

---

> ### Author Rebuttal · Authors · 2026-03-31
>
> We sincerely thank the reviewer for the thorough review and valuable suggestions. We hope the following responses can address your concerns.
>
> **W: Evaluation on high-resolution and fine-grained benchmarks.**
>
> We agree that relying solely on UCF-101 has limitations. To evaluate generalization to diverse, high-resolution content, we conducted new experiments on the DAVIS 2019 test set [1] at 720p. DAVIS features complex in-the-wild videos with fast motion, occlusions, and fine details. As shown in the table below, Flash-VAED demonstrates superior performance on both Wan and LTX backbones, significantly outperforming the baselines across all the metrics.
> | **Model** | **PSNR  ↑** | **SSIM ↑** | **LPIPS ↓** |
> | :--- | :---: | :---: | :---: |
> | Wan 2.1 | 34.13 | 0.9079 | 0.0487 |
> | LightVAE-Wan 2.1 | 27.92 | 0.8397 | 0.2069 |
> | Flash-VAED-Wan 2.1  | **32.03** | **0.8825** | **0.0769** |
> | LTX  | 29.36 | 0.8134 | 0.1555 |
> | Turbo-VAED-LTX | 26.82 | 0.7805 | 0.1783 |
> | Flash-VAED-LTX | **28.26** | **0.8031** | **0.1214** |
>
> **Q1: VBench-2.0 configurations.**
>
> To ensure fair comparisons, we strictly followed the officially recommended inference settings for each baseline: 480x832 resolution with 81 frames for Self Forcing, and 448x832 resolution with 61 frames for FastVideo.
>
> **Q2: Resolution generalization.**
>
> First, we clarify that the Wan-1.3B model evaluated in our paper natively targets only 480p inference. Nevertheless, as demonstrated by our new 720p DAVIS experiments above, Flash-VAED generalizes excellently to higher resolutions. This zero-shot spatial generalization stems from its fully convolutional architecture. The decoder's local receptive fields learn shift-invariant mappings from latent representations to pixel space. Since low-level visual statistics (e.g., textures, edges) remain consistent across spatial scales, the learned kernels effectively reconstruct larger latent grids.
>
> **Q3:**
>
> To ensure full reproducibility, we clarify that the specific variant utilized in all related experiments is Wan 2.1 T2V-1.3B.
>
> **Q4:**
>
> First, we confirm that temporal cache is a highly effective technique that enables the VAE decoder to decode high-resolution, long videos with low VRAM overhead. Therefore, we preserved this design. We employ the cache in the 3D convolution part of Flash-VAED to capture temporal dependencies. However, we omit it in the 2D layers, as 2D convolutions process each frame independently and do not require cross-frame temporal interactions.
>
> **Limitations, Failure Cases, and Architectural Generalization.**
>
> We sincerely thank the reviewer for providing these insightful questions, which prompted us to further reflect on the limitations and broader generalization of our proposed method.
>
> First, we agree that a Failure Case Analysis is essential for a comprehensive evaluation. Although Flash-VAED maintains high reconstruction quality alongside significant acceleration, we observed that performance may decline in extreme scenarios involving a large number of rapidly moving tiny objects or human faces. These cases represent the current boundaries of our model's representational capacity, where the highly efficient architecture may struggle to capture dense, high-frequency temporal changes. We will include a dedicated section with visual examples of these failure cases in the revised manuscript to provide a more balanced view of the method's performance.
>
> Second, regarding the generalizability of our approach across diverse architectures, we offer the following perspectives:
>
> 1. Pruning Strategy: In Flash-VAED, the high-resolution layers are processed using 2D convolutions, which are inherently non-causal. Since our pruning is executed after this operator replacement, our experiments on Wan and LTX already indicate that the pruning criterion remains effective for non-causal structures. For Transformer-based VAE decoders, this channel pruning strategy may be adapted to Feed-Forward Network (FFN) and embedding dimensions, which often exhibit structural redundancy in other generative tasks.
>
> 2. Operator Replacement: For non-causal, convolution-based VAE decoders, replacing standard convolutions with depthwise separable convolutions remains a viable method for reducing parameter counts. For Transformer-based VAE decoders, where convolutions are no longer the dominant operators, the focus of operator replacement may shift toward accelerating the attention module by exploring efficient alternatives such as linear attention.
>
> While the specific operators differ across backbones, the underlying approach of identifying and replacing heavy, redundant components with efficient counterparts remains a promising direction. We will include a more nuanced discussion of these architectural adaptations in the revised manuscript.
>
> **Reference**
>
> [1] The 2019 DAVIS Challenge on VOS. arXiv:1905.00737, 2019.

---

> > ### Author Rebuttal · Reviewer_mwsW · 2026-04-01
> >
> > I thank the authors for their detailed rebuttal and the additional experiments provided.
> >
> > The clarifications and new data effectively address my concerns. In particular, the inclusion of the DAVIS dataset results strengthens the empirical evidence supporting the proposed method. I strongly recommend that the authors incorporate these new results with throughout metrics (e.g. FPS) as in Tab. 1 into the next revision to ensure the findings are persuasive to the broader community.
> >
> > I maintain my positive assessment of the work and keep my recommendation as 5-Accept.

---

### Official Review · Reviewer_ibSY · 2026-03-11

**Soundness:** 3
**Presentation:** 3
**Significance:** 3
**Originality:** 2
**Overall Recommendation:** 4
**Confidence:** 4

**Summary:**

This paper studies an increasingly relevant bottleneck in modern video generation: once the denoising backbone is accelerated, VAE decoding can become a substantial fraction of total inference time. The paper proposes Flash-VAED, a plug-and-play decoder transfer framework for speeding up pretrained video VAE decoders without retraining the full video generation model. The method combines three main ideas: channel pruning based on redundancy and reconstructability, stage-wise replacement of expensive operators in the decoder, and a multi-stage distillation procedure to preserve compatibility with the original latent space. The paper evaluates the approach on two modern video VAE backbones, Wan and LTX-Video, and reports substantial decoder speedups while maintaining good reconstruction quality. It also shows end-to-end gains when the accelerated decoder is inserted into already-optimized video generation pipelines.

**Compliance With Llm Reviewing Policy:**

Affirmed.

**Final Justification:**

The paper addresses a practical and increasingly important systems bottleneck in video generation by accelerating pretrained video VAE decoders through a modular transfer framework, and I view its strongest contributions as solid empirical utility, sensible design choices, and clear end-to-end relevance. The authors’ rebuttal was helpful and did address several of my main concerns, particularly by providing stronger plug-and-play evidence, a stronger pruning baseline, and preliminary discussion of additional backbones, which increases my confidence in the work’s soundness and practical value. However, because the broader generalization claims and the full breadth of comparative validation are still only partially established, I maintain my original overall assessment: this is a useful and technically solid contribution with moderate originality, but not enough evidence was added to change my score.

**Key Questions For Authors:**

1.The paper makes a strong plug-and-play compatibility claim. What additional quantitative evidence can the authors provide to show that the transferred decoder remains wel aligned with the original latent space beyond the specific pipelines tested here? A stronger answer would improve my confidence in the soundness and generality of the method.

2.Can the authors provide more complete comparisons against the closest decoder-transfer baselines across all applicable backbones? A more uniform comparison would help clarify both the significance and the actual empircal advantage of the proposed approach.

3.Can the authors compare their prunig strategy against stronger structured pruning baselines rather than only random pruning? If the proposed criterion remains clearlybetter, that would strengthen my view of the method’s technical contribution.

**Limitations:**

No. The paper would benefit from a clearer discussion of technical limitations, especially the extent of its generalization beyond the tested backbones, the strength of the evidence for plug-and-play compatibility, and the practical cost of transferring the decoder.

**Strengths And Weaknesses:**

The paper has several clear strengths. First, it focuses on a practical and timely problem. A lot of recent work has concentrated on accelerating the denoising model, and it is useful to point out that decoder latency then becomes a meaningful bottleneck. The paper makes this motivation concrete with latency breakdowns rath er than relying only on intuition.

Second, the method is fairly well designed. The pruning idea is more thoughtful than simple channel removal, sin ce it tries to account for redundancy while respecti ng the residual structure of the decoder. The stage-wise operator replacement is also sensible, since different decoder stages play different roles and should not necessarily be optimized in the same way. The overall method is modular and easy to understand at a high level.

Third, the empirical results are meaningful from a systemss perspective. The paper evaluates on two distinct video VAE families and shows gains not only in isolated decoder benchmarks but alsso in end-2-end generation pipelines. That makes the contribution more convincing than a purely ccomponent-level optimization result. The inclusion of experiments on both high-end and edge  oriented hardware is also a practical plus.

That said, I have a few importa nt reservations The biggest is that some of the broadest claims are not fully supported by the current evidence. The paper argues that the transferred decoder remains aligned with the original latent space and can serve as a plug-and-play replacement, but the evidence for this is sommewhat limited. The compatibility results on the tested pipelines are encouraging, but they do not fully establish that the method  will be robust across a broader range of pipelines, tasks, or decoder settings.

A second weakness is that the comparison to related work could be stronger and more symmetric. The paper discusses strong recent baselines, but the main experimental tables do not always compare all relevant methods across all applicable backbones. Because this is a crowded and fast-moving area, the paper would be stronger if it gave a more direct and uniform comparison against the most relevant concurrent decoder-transfer approaches.

Third, the ablation evidence for some core design choices is not as strong as it could be. In particular, the pruning analysis mainly compares the proposedpruning strategy against random pruning. That is too weak a baseline for such a central component. A stronger comparison against standard structured pruning or channel saliency methods would make the technical case more convincing.

Fourth, although the method isprctically useful, the originality is moderate rather than high. The most distinctive parts are the specific pruning formulation and the stagewise transfer recipe. However, the overall direction of decoder transfer and decoder distillation for efficient video generation is already being actively explored by closely related recent work. So the paper feels more like a strong engineering and design contribution than a major conceptual advance.

---

> ### Author Rebuttal · Authors · 2026-03-31
>
> We sincerely thank the reviewer for the thorough review and valuable suggestions. We hope the following responses can address your concerns.
>
> **W1 & Q1: Deployment on additional baselines to verify plug-and-play capability.**
>
> We agree that verifying Flash-VAED across broader pipelines is essential to validate its plug-and-play capability and latent space alignment.
>
> To provide stronger quantitative evidence, we deployed Flash-VAED on Causal Forcing [1], a recent representative method in autoregressive few-step distillation. We evaluated the generation performance using VBench 2.0. Due to space constraints, we report a selected subset of metrics below (abbreviated by initials).
>
> As shown in the table, when replacing the original decoder with Flash-VAED, the performance closely matches the original Wan decoder across all dimensions. In stark contrast, LightVAE suffers from severe degradation.
>
> These results validate that Flash-VAED exhibits robust generalization and plug-and-play capabilities, remaining well-aligned with the original latent space even in entirely new generative pipelines. We will include additional visual demonstrations in the revised manuscript to further substantiate these findings.
>
> | Model | DSR ↑| HC ↑| HId ↑| HIn ↑| IR ↑| MOU ↑| MVC ↑|
> | :--- | :---: | :---: | :---: | :---: | :---: | :---: | :---: |
> | Wan 2.1 | 0.217 | 0.986 | 0.785 | 0.6 | 0.924 | 0.131 | 0.424 |
> | LightVAE-Wan 2.1 | 0.198 | 0.953 | 0.527 | 0.57 | 0.819 | 0.121 | 0.072|
> | Flash-VAED-Wan 2.1 | **0.237** | **0.986** | **0.741** | **0.59** | **0.918** | **0.141** | **0.318**|
>
> **W2 and Q2: Generalization to additional backbones.**
>
> We sincerely thank the reviewer for this valuable suggestion. We agree that validating our approach across a wider range of backbones would further substantiate our method's generalization capabilities. To address this, we have actively begun validating our algorithm on the Hunyuan Video [2] and CogVideoX [3] VAE decoders.
>
> First, by analyzing their architectures, we observed they share both the computational bottlenecks (massive channel counts and frequent CausalConv3D usage) and the structural characteristics (cascaded residual blocks and temporal upsampling concentrated in low-resolution layers) with Wan and LTX. This fundamental homogeneity provides a solid theoretical basis for the transferability of our method.
>
> Second, our preliminary experiments on these backbones yielded promising results. Applying SVD to the channel output feature maps of randomly sampled layers revealed that retaining merely 12.5% and 11.72% of singular values preserves 99% of the total energy, supporting the low-rank assumption behind our channel pruning. Furthermore, our operator optimization, which replaces 3D convolutions with 3D depthwise separable convolutions in low-resolution layers and 2D convolutions in high-resolution layers, successfully reduced the parameter count to ~1/10 of the original size and achieved a ~35% inference speedup.
> Due to the extremely tight time constraints, we have not yet completed the full implementation and validation experiments for these new backbones. However, if this paper is accepted, we will provide more related results in the revised manuscript.
>
> **W3 and Q3: Comparison against stronger structured pruning baselines.**
>
> We appreciate the reviewer’s suggestion to include stronger structured pruning baselines. To further validate our pruning strategy, we compared our method against $L_1$-norm based pruning, a classic and robust structured pruning criterion. In this baseline, channels are ranked and pruned based on the sum of the absolute values of their weights ($L_1$-norm), assuming that filters with smaller weight magnitudes are less important.
>
> As shown in the table below, our proposed pruning strategy consistently outperforms both Random and $L_1$-norm based pruning. Specifically, our method achieves a PSNR of 32.27 dB, representing a substantial gain of 1.0 dB over L1-norm based pruning and over 4.0 dB over random pruning. Significant improvements are also observed in SSIM and LPIPS. These results demonstrate that our criterion is more effective than traditional weight-magnitude-based methods at identifying redundant channels in VAE decoders.
>
> | Pruning Method | PSNR (dB) ↑ | SSIM ↑ | LPIPS ↓ |
> | :--- | :---: | :---: | :---: |
> | Random Pruning | 28.21 | 0.8786 | 0.1359 |
> | L1-Norm Based Pruning | 31.27 | 0.9277 | 0.0686 |
> | **Ours** | **32.27** | **0.9397** | **0.0558** |
>
> **Reference**
>
> [1] Causal Forcing: Autoregressive Diffusion Distillation Done Right for High-Quality Real-Time Interactive Video Generation. arXiv preprint arXiv:2602.02214, 2026.
>
> [2] Hunyuanvideo: A systematic framework for large video generative models. arXiv preprint arXiv:2412.03603, 2024.
>
> [3] CogVideoX: Text-to-Video Diffusion Models with An Expert Transformer. arXiv preprint arXiv:2408.06072, 2024.

---

> > ### Author Rebuttal · Reviewer_ibSY · 2026-04-04
> >
> > Thank you for your response. The rebuttal addresses several of my main concerns in a meaningful way, particularly by adding stronger evidence for plug-and-play compatibility, broader preliminary validation on additional backbones, and a more appropriate pruning baseline comparison. These additions increase my confidence in the practical value and technical soundness of the work, although some questions around the breadth of validation and generalization remain only partially resolved.

---

### Official Review · Reviewer_qSEH · 2026-03-12

**Soundness:** 3
**Presentation:** 3
**Significance:** 4
**Originality:** 3
**Overall Recommendation:** 4
**Confidence:** 5

**Summary:**

This paper introduces Flash-VAED, a plug-and-play acceleration framework for video VAE decoders. It achieves approximately 6x acceleration across two representative video decoders from Wan 2.1 and LTX-Video. To realize this, the framework employs three core strategies: (1) an independence-aware channel pruning technique, retaining only a fraction of the channels while linearly reconstructing the rest; (2) a stage-wise dominant operator optimization that replaces computationally expensive CausalConv3D operations with efficient alternatives like 3D depthwise separable convolutions in deep layers and 2D convolutions in shallow layers; and (3) a three-phase dynamic training pipeline. Extensive experiments show that Flash-VAED significantly accelerates both the VAE decoding and the end-to-end video generation pipeline, while outperforming the two compared baselines, LightVAE-Wan 2.1 and Turbo-VAED-LTX.

**Compliance With Llm Reviewing Policy:**

Affirmed.

**Final Justification:**

I have read the rebuttal and my concerns have been addressed. I will keep my score.

**Key Questions For Authors:**

1. Can the authors provide more comprehensive temporal consistency evaluations beyond standard metrics (PSNR/SSIM/LPIPS), to definitively assess whether the proposed method disrupts temporal coherence?
2. Since this paper targets overall 'efficiency' rather than just acceleration, could the authors provide a more detailed efficiency comparison beyond runtime, specifically regarding parameter counts and computational costs? Furthermore, exactly how much redundancy is reduced at each individual pruning stage?
3. The visual results in Fig. 8 suggest that LightVAE might not be fully plug-and-play. For the Wan 2.1 decoder, another highly representative and widely used plug-and-play alternative is TAE-Wan2.1. Can the authors provide a comparison between it and Flash-VAED?

**Limitations:**

No. See weaknesses above.

**Strengths And Weaknesses:**

### Strengths

1. The paper achieves significant acceleration on two highly representative decoders: Wan 2.1 and LTX-Video. Given the increasing latency bottleneck of VAE decoders of recent DiT-only acceleration works, as well as the growing demand for real-time video generation and fast previews, this work provides a substantial contribution to the video diffusion field.
2. The analysis of VAE channel redundancy, module-specific computational bottlenecks, and post-pruning performance variations are all rigorous. This ensures the proposed method to be built upon a solid theoretical foundation.
3. The paper is well-structured, and three proposed ideas are easy to follow.
4. The training process requires only 10K samples and approximately 300 GPU-hours on 4 GPUs, which is  relatively resource-efficient and highly reproducible.


### Weaknesses

1. The visual demonstrations (Fig. 1, Fig. 8) are limited to relatively small resolutions and simple scenes. This hinders the ability to accurately judge whether Flash-VAED can effectively reconstruct fine-grained, realistic details, and don't degrade severely on complex elements like face and small text.
2. The metrics are limited to PSNR/SSIM/LPIPS. Although the authors provide VBench metrics for the overall video generation pipeline , there is still a lack of clear evidence demonstrating that the method does not severely degrade temporal smoothness and consistency, especially since 3D operations are replaced by purely spatial 2D convolutions in the high-resolution layers.
3. The quantitative performance gain over the compared baselines might partially stem from the absence of a GAN loss in the proposed training objective. Empirically, omitting a GAN loss in decoder training often leads to artificially higher consistency metrics but can cause over-smoothing or visual artifacts. Compounded by weakness 1, it is difficult to confidently determine if this introduces negative visual impacts.

---

> ### Author Rebuttal · Authors · 2026-03-31
>
> We sincerely thank the reviewer for the thorough review and valuable suggestions. We hope the following responses can address your concerns.
>
> **W1 and W3: Evaluation on high-resolution and fine-grained benchmarks.**
>
> To address the reviewer’s concerns about high-resolution and fine-grained details, we evaluated Flash-VAED on the 720p DAVIS 2019 test set [1], known for complex motions and intricate textures. As shown below, Flash-VAED significantly outperforms baselines across Wan and LTX backbones.
>
> While omitting GAN loss may cause over-smoothing, this typically happens under purely pixel-level losses (e.g., L1). To preserve quality, we incorporate SSIM and LPIPS constraints. LPIPS enforces deep-feature clarity, effectively mitigating over-smoothing without adversarial training. Consequently, our strong PSNR confirms strict fidelity to high-frequency details, and LPIPS scores verify high perceptual sharpness.
> | Model | PSNR ↑| SSIM ↑| LPIPS ↓| Warping Error ↓|
> | :--- | :---: | :---: | :---: | :---: |
> | Wan | 34.13 | 0.9079 | 0.0487 | 0.017783 |
> | LightVAE | 27.92 | 0.8397 | 0.2069 | 0.026640 |
> | TAE | 29.64 | 0.8617 | 0.1076 | 0.020801 |
> | Flash-VAED | **32.03** | **0.8825** | **0.0769** | **0.018662** |
> | LTX | 29.36 | 0.8134 | 0.1555 | 0.022071 |
> | Turbo-VAED | 26.82 | 0.7805 | 0.1783 | 0.026261 |
> | Flash-VAED | **28.26** | **0.8031** | **0.1214** | **0.024267** |
>
> **W2 and Q1: Quantitative assessment of temporal consistency and smoothness.**
>
> We sincerely thank the reviewer for this insightful comment. We agree that comprehensively evaluating temporal consistency is crucial, especially given our replacement of 3D convolutions with 2D convolutions in high-resolution layers.
>
> While VBench 2.0 metrics (e.g., Human Clothes) used in our evaluation reflect temporal smoothness to some extent, we further adopt the Warping Error [3] for a more rigorous assessment. Widely used in video synthesis, it measures pixel-level deviation between consecutive frames aligned by optical flow.
>
> For a given video sequence, let $x_t$ be the ground truth frame at time $t$, and $\hat{x}_t$ be the reconstructed frame. We first extract the optical flow $F_t$ from the ground truth frames using a pre-trained RAFT model:
>
> $$F_t = \text{RAFT}(x_t, x_{t-1})$$
>
> The Warping Error is then computed as the average $L_1$ distance between the current reconstructed frame and the previous reconstructed frame warped by the ground truth flow:
>
> $$
> E\_{warp} = \frac{1}{T-1} \sum\_{t=2}^{T} \lVert \hat{x}\_t - \text{Warp}(\hat{x}\_{t-1}, F\_t) \rVert\_1
> $$
>
> As shown in the table above, Flash-VAED maintains a Warping Error highly comparable to the original backbones. This demonstrates that our method preserves the original VAE decoder's temporal consistency while significantly accelerating inference, validating the feasibility of using 2D convolutions in high-resolution layers.
>
> **Q2: Comprehensive efficiency and model lightweighting analysis.**
>
> We sincerely thank the reviewer for pointing this out. We agree our method targets comprehensive lightweight design beyond mere inference acceleration. Using the Wan VAE decoder as an example, we supplement evaluations on parameter count, peak inference VRAM, and specific gains from each lightweighting step, as shown in the table below.
>
> As shown, both Flash-VAED and LightVAE compress the original parameters to less than 1/10 and reduce peak VRAM to 2.00 GB or less. However, unlike LightVAE, which suffers severe quality degradation, Flash-VAED achieves this extreme lightweighting while preserving the original generation quality.
>
> Regarding individual contributions, the table illustrates the redundancy reduced at each stage. Operator optimization ("Wan + Operator") significantly decreases both parameter count and computational overhead. Conversely, channel pruning ("Wan + Prune") specifically targets peak VRAM reduction, drastically dropping peak VRAM from 7.11 GB to 2.31 GB.
> | Model | Number of Parameters ↓| Peak VRAM ↓|
> | :--- | :---: | :---: |
> | Wan | 73.30 M | 7.11 GB |
> | Wan + Prune | 65.20 M | 2.31 GB |
> | Wan + Operator | 5.63 M | 3.51 GB |
> | LightVAE | 4.62 M | 1.17 GB |
> | TAE | 11.32 M | 17.01 GB |
> | Flash-VAED | 5.86 M | 2.00 GB |
>
> **Q3: Comparison against TAE-Wan2.1.**
>
> We sincerely apologize for omitting this baseline. Since the initial TAE project[2] focused exclusively on the Hunyuan VAE decoder, we unfortunately missed its subsequent extension to the Wan 2.1 decoder, which we consider concurrent work.
> Following the reviewer's advice, we have conducted comparative experiments with TAE-Wan2.1. As shown in the comprehensive comparison tables provided above, Flash-VAED-Wan 2.1 significantly outperforms TAE-Wan2.1 across all evaluated dimensions, including visual quality, temporal consistency, and model lightweighting.
>
> **Reference**
>
> [1] The 2019 DAVIS Challenge on VOS. arXiv:1905.00737, 2019.
>
> [2] TAEHV: Tiny AutoEncoder for Hunyuan Video
>
> [3] Learning Blind Video Temporal Consistency. ECCV2018.

---

> > ### Author Rebuttal · Reviewer_qSEH · 2026-04-03
> >
> > I have read the rebuttal and my concerns have been addressed. I will keep my score.

---

### Decision · Program_Chairs · 2026-04-30

**Decision:**

Accept (regular)

**Comment:**

This paper initially received positive feedback, with reviewers praising its focus on a timely and practical problem (VAE decoder latency), its good methodology, and its rigorous analysis that yielded acceleration on representative models. During the rebuttal phase, the authors provided valuable additional details that were well-received by the reviewers. These included verification on the high-resolution DAVIS 2019 dataset, empirical evidence demonstrating plug-and-play compatibility, preliminary validation on additional backbones (Hunyuan Video and CogVideoX), and a more robust pruning baseline comparison.

Following the rebuttal, the reviewers indicated that the major issues had been adequately addressed, resulting in universally favorable ratings. The remaining reservations stem from one reviewer regarding the breadth of validation across diverse datasets and the full extent of the method's generalization to other video VAE architectures. While the authors supplied new evidence on the DAVIS 2019 dataset and preliminary experiments on additional backbones (limited by time constraints), this reviewer felt that the universal "plug-and-play" generalization claim is not yet conclusively proven, though they still recommend acceptance.

The AC agrees with the overall acceptance rating. Given the preliminary results provided in the rebuttal, the AC does not view the generalization concern as a critical flaw that precludes acceptance. However, the AC urges the authors to appropriately scope and revise their generalization claims in the final text to better reflect the current evidence. Furthermore, the authors must incorporate the new results presented during the rebuttal into the revised manuscript, specifically the evaluations on Causal Forcing, the Hunyuan Video backbone, and the high-resolution DAVIS 2019 dataset.